# Phage production is blocked in the adherent-invasive *Escherichia coli* LF82 upon macrophage infection

Pauline Misson[1], Emma Bruder[2], Jeffrey K. Cornuault[1], Marianne De Paepe[1], Pierre Nicolas[3], Gaëlle Demarre[2], Goran Lakisic[1], Marie-Agnès Petit[1], Olivier Espeli[2], François Lecointe[1]*

1 Université Paris-Saclay, INRAE, AgroParisTech, Micalis Institute, Jouy-en-Josas, France, 2 Center for Interdisciplinary Research in Biology (CIRB), College of France, CNRS, INSERM, Université PSL, Paris, France, 3 Université Paris-Saclay, INRAE, MaIAGE, Jouy-en-Josas, France

* francois.lecointe@inrae.fr

**Data Availability Statement:** Genome and reads submissions. The re-annotated genomes of the phages are available from the European Nucleotide Archive browser (http://www.ebi.ac.uk/ena/

## Abstract

Adherent-invasive *Escherichia coli* (AIEC) strains are frequently recovered from stools of patients with dysbiotic microbiota. They have remarkable properties of adherence to the intestinal epithelium, and survive better than other *E. coli* in macrophages. The best studied of these AIEC is probably strain LF82, which was isolated from a Crohn's disease patient. This strain contains five complete prophages, which have not been studied until now. We undertook their analysis, both *in vitro* and inside macrophages, and show that all of them form virions. The Gally prophage is by far the most active, generating spontaneously over $10^8$ viral particles per mL of culture supernatants *in vitro*, more than 100-fold higher than the other phages. Gally is also over-induced after a genotoxic stress generated by ciprofloxacin and trimethoprim. However, upon macrophage infection, a genotoxic environment, this over-induction is not observed. Analysis of the transcriptome and key steps of its lytic cycle in macrophages suggests that the excision of the Gally prophage continues to be repressed in macrophages. We conclude that strain LF82 has evolved an efficient way to block the lytic cycle of its most active prophage upon macrophage infection, which may participate to its good survival in macrophages.

## Author summary

Prophages are bacterial viruses stably integrated into their host, to which they can provide new functions, thus increasing their fitness in the environment. Thereby, they can participate to the virulence of bacterial pathogens. However, prophages are double-edged swords that can be awakened in response to genotoxic stresses, resulting in the death of their bacterial host. This raises the question of the effect of this type of stress in the natural environments where their bacterial hosts exert their virulence. In this study, we characterized the five active prophages present in *Escherichia coli* LF82, a strain belonging to the intestinal microbiota and suspected to be involved in Crohn's disease *via* its ability to invade

browser/view) with the following accession numbers: OV696608 for Gally, OV696612 for Perceval, OV696610 for Tritos, OV696611 for Cartapus and OV696614 for Cyrano. Raw data obtained from the virome sequencing have been deposited (accession number: ERR8973296).

**Funding:** This work was supported by the Agence Nationale de la Recherche https://anr.fr (Persist3R contract, reference: ANR-18-CE35-0007, Grant recipient: M-A. P.). P.M. was supported by a fellowship from the French Ministère de l'Enseignement Supérieur et de la Recherche and from the MICA division of INRAE. E.B. was supported by a fellowship from the French Ministère de l'Enseignement Supérieur et de la Recherche. The funders had no role in study design, data collection and analysis, decision to publish, or preparation of the manuscript.

**Competing interests:** The authors have declared that no competing interests exist.

macrophages, a highly genotoxic environment. We show that LF82 inhibits the awakening of its prophages in macrophages, allowing it to survive there. Moreover, deletion of its most active prophage does not affect the viability of LF82 in this environment. These results suggest that LF82 has tamed its prophages in macrophages and also suggest that if they convey fitness advantages, they probably do so in environments differing from macrophages, and which remain to be discovered.

## Introduction

Lysogens, the bacteria hosting functional prophages (either integrated into their genome or as freely replicating episomes), are abundant in natural ecosystems, representing approximately half of all completely sequenced strains [1], and up to 70% of intestinal bacteria [2–5]. Lysogeny is often considered as beneficial for the bacterium, thanks to the expression of prophage genes named "morons". Morons are genes that are autonomously expressed (*i.e.* not under the control of the lysogeny master regulator), and are not involved in the lytic cycle of the phage but provide the bacterial host with some gain of function (for a review see [6]). Moron functions are diverse, ranging from protection against infection by other phages [7–9], metabolic genes [2,10] or adaptation to a given environment [11,12], and many remain to be characterized. However, this potential benefit of lysogeny is counterbalanced by the permanent danger of lysis, due to prophage induction, *i.e.* its entry into a lytic cycle upon derepression of the lysogeny master regulator. Bacterial growth may be hindered for lysogens compared to non-lysogens if the prophage is constantly induced in a significant proportion of the population. This burden is important in the case of an *Escherichia coli* lysogen propagated in the gastro-intestinal tract (GIT) of monoxenic mice [13], and has also been observed for *Lactobacillus reuterii* lysogens, during transit in conventional mice [3]. In both cases, prophage induction was RecA-dependent, suggesting that the inducing signal was due to genotoxic stress. In the mammalian gut, molecules activating the SOS response, might not only be produced by the host itself, but also by the microbiota [14]: for *L. reuterii*, fructose consumption and acetic acid release was suggested as the origin of the genotoxic stress inducing its prophages in the GIT [3]. Few RecA-independent pathways have also been described for prophage induction [15–19] and other sensors and pathways certainly await discovery.

Characterizing lysogeny, and the signals regulating prophage induction in natural settings, is therefore critical for the understanding of bacterial behaviors in their natural environments, and more particularly those of bacterial pathogens. Indeed, bacterial pathogens are often lysogens, and even poly-lysogens [1]. The first prophage morons to be described were virulence factors, such as the shiga-toxin, to mention just one of many important phage-encoded toxins [6]. All happens as if prophages were adjustment variables, allowing pathogens to rapidly adapt to new niches and compete with the local inhabitants [20]. The adherent-invasive *Escherichia coli* (AIEC) genomes are also richly decorated with prophages, and strain LF82, its best characterized member, encodes five prophages predicted to be complete and functional, based on genomic analyses [21,22]. One of these, named "prophage 1", was even reported as significantly associated to the *E. coli* strains isolated from Crohn's disease patients [23]. Yet, whether these prophages are functional (able to complete lytic cycles, form virions and multiply), and what kind of signal or stress induces them, had not been investigated.

AIEC have two remarkable properties: they adhere to the intestinal epithelium [24], and they invade and multiply in macrophages [25,26], being able to form biofilm within vacuoles [27].

Interestingly, it was recently established that the macrophage environment can provoke Lambda prophage induction by 26-fold compared to its spontaneous induction *in vitro* from a laboratory model *E. coli* strain [16]. This high induction level, which leads *in vitro* to the lysis of ~90% of the bacteria, probably facilitates the clearing work of macrophages [16]. In the vacuolar compartment, Lambda induction was not dependent on RecA, as observed *in vitro*, but rather on PhoP, a DNA-binding protein belonging to a two-component system with its sensor PhoQ. This system is involved in the adaptation of bacteria to magnesium-poor environments and in resistance to acid stress and antimicrobial peptides [28]. However, the exact mechanism involving PhoP in Lambda induction remains unknown to date. An anti-microbial peptide, mCramp1, was suggested to be at the origin of this induction, *via* a bacterial membrane stress [16]. Whether prophages from natural *E. coli* isolates are also induced and provoke bacterial lysis in macrophages remains unknown. This question is of importance for LF82 that survives in macrophages and contains five putative active prophages.

To further understand the good survival of LF82 in macrophages, we undertook the characterization of its prophages, both *in vitro* and inside macrophages. We show that the five prophages form virions *in vitro*. One of them, the phage Gally (formerly prophage 1) dominates the culture supernatants: its spontaneous induction level is high enough to generate above $10^8$ particles per mL of culture during exponential growth in rich medium. Interestingly, and contrary to expectations, Gally was not induced upon growth in macrophages. We hypothesize that the remarkable survival of LF82 in macrophage is partly due to its ability to control the induction level of its most active prophage.

## Results

### The prophages of LF82 encode various morons

Previous analysis of the *E. coli* LF82 genome had identified four putative complete prophages and one plasmid, pLF82, subsequently found to be homologous to the *Salmonella* SSU5 phage [21]. Our analysis of the *E. coli* LF82 genome did not detect any other complete or degraded prophage using PHASTER. Here we named the five prophages Gally (previously prophage 1), Perceval (prophage 2), Tritos (prophage 3), Cartapus (prophage 4) and Cyrano (pLF82), and upon updated re-annotation, these were introduced into the European Nucleotide Archive database (see Material and Methods). The phage-plasmid Cyrano closely resembles the phage SSU5 (81% identity, 65% coverage, see S1 Fig panel A), and is predicted to have a siphovirus morphology. Among integrated prophages, Gally is partially homologous to the podophage HK620 (97% identity on 44% coverage, Fig 1 top, [29]), and follows the general genetic organization of P22 (Lederbergvirus genus, [30]). Perceval is a close relative of Ev207, a phage isolated from the infant gut and homologous to Lambda [2]. Tritos is also distantly related to Lambda, and close to another infant gut phage, named Ev081 [2]. Finally, Cartapus is a P2-related phage, closely similar to Fels2 (Felsduovirus genus, [30]).

To identify the morons of the five prophages, transcriptomes of LF82 grown *in vitro* in LB (2 repeats) were analyzed (S1 Table) [27]. Morons were identified as genes different from the master regulator or other typical phage genes and transcribed at least 5-fold above the local transcriptional background of the prophage region (see Material and Methods, Fig 1). Cyrano was the richest in moron content (13 genes), followed by Perceval and Tritos (7 genes), then Cartapus (2 genes), while Gally was apparently devoid of any moron (Table 1).

Among the identified morons, Perceval was found to encode, the *sitABCD* operon, a virulence factor allowing improved manganese and iron import. Tritos and Cyrano encoded several toxins, sometimes next to a recognized or predicted anti-toxin (HokD for Tritos, KacT-CYRAN27, RelE and HicA-HicB for Cyrano). We conclude that all prophages but Gally encode morons.

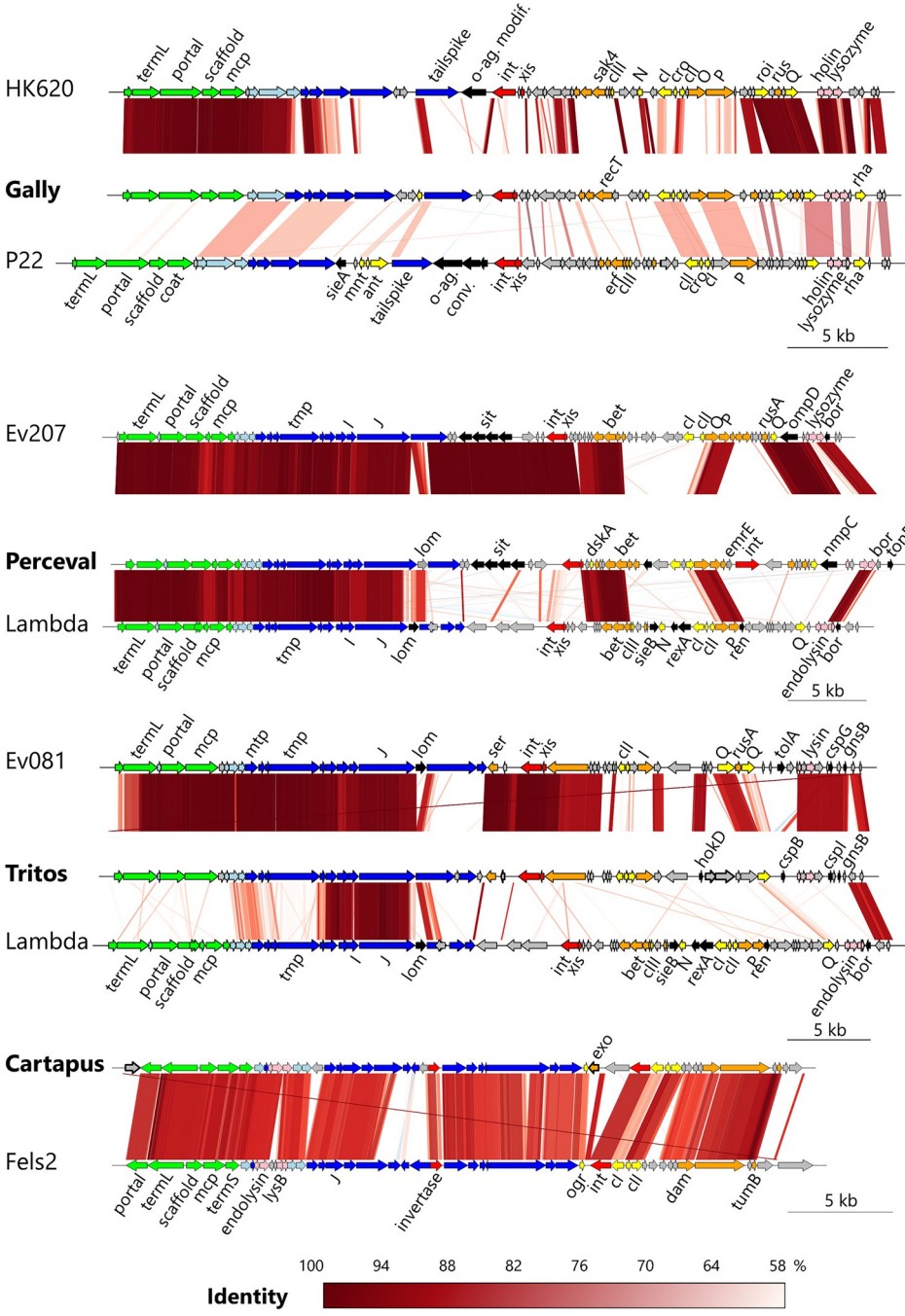

**Fig 1. Whole genome comparisons of the four integrated prophages.** Each LF82 phage (in bold) is compared with tBLASTx to its closest relative in databases (up), as well as to an ICTV classified prototype (down), and displayed using the R Genoplot package. Gene color code: green, capsid; light blue, connector; dark blue, tail; red, integration and excision; orange, DNA metabolism; yellow, transcriptional regulators; pink, lysis; grey, hypothetical. Morons identified by our transcriptomic analysis are shown in black (predicted function) or in arrows with thick black border (unknown or putative function).

**Table 1. Prophage genes expressed at least 5-fold above local background in strain LF82 grown in LB medium (overnight culture).**

| Prophage name | Locus tag | Annotation[a] | Fold above BackG[b] |
|---|---|---|---|
| *Gally* | *LF82_107* | *Repressor* | *47* |
| *Perceval* | *LF82_154* | *CI repressor* | *11* |
| Perceval | LF82_152 | transmembrane protein | 81 |
| Perceval | LF82_1505 | Outer membrane porin NmpC | 10 |
| Perceval | LF82_160 | TonB fragment | 8 |
| Perceval | LF82_184 | SitD iron/manganese transport protein | 5 |
| Perceval | LF82_185 | SitC iron/manganese transport protein | 4[c] |
| Perceval | LF82_186 | SitB iron/manganese transport protein | 10 |
| Perceval | LF82_187 | SitA iron/manganese transport protein | 13 |
| *Tritos* | *LF82_287* | *CI Repressor* | *24* |
| Tritos | LF82_2833 | DNA binding transcriptional regulator | 8 |
| Tritos | LF82_0907 | Uncharacterized protein YnfN | 27 |
| Tritos | LF82_0377 | Cold shock-like protein CspI | 93 |
| Tritos | LF82_0371 | Cold shock-like protein CspB | 59 |
| Tritos | LF82_280 | ImmA/IrrE metallo-endopeptidase | 14 |
| Tritos | LF82_281 | hypothetical protein | 30 |
| Tritos | LF82_1023 | HokD toxin | 3792 |
| *Cartapus* | *LF82_413* | *CI repressor protein* | *5* |
| Cartapus | LF82_783 | putative exonuclease protein | 15 |
| Cartapus | LF82_789 | hypothetical protein | 7 |
| *Cyrano* | *CYRAN_45* | *putative repressor* | *10* |
| Cyrano | CYRAN_26 | KacT Acetyltransferase-type toxin | 35 |
| Cyrano | CYRAN_27 | putative antitoxin | 60 |
| Cyrano | CYRAN_29 | RelE/ParE putative toxin | 9 |
| Cyrano | CYRAN_30 | Hypothetical protein | 6 |
| Cyrano | CYRAN_40 | Hypothetical protein | 15 |
| Cyrano | CYRAN_43 | Hypothetical protein | 723 |
| Cyrano | CYRAN_50 | putative antitoxin HicB | 14 |
| Cyrano | CYRAN_51 | HicA toxin protein | 30 |
| Cyrano | CYRAN_81 | Hypothetical protein | 12 |
| Cyrano | CYRAN_99 | Septum site-determining protein | 9 |
| Cyrano | CYRAN_102 | Hypothetical protein containing coiled-coil | 6 |
| Cyrano | CYRAN_109 | Hypothetical protein | 6 |
| Cyrano | CYRAN_117 | Hypothetical protein | 12 |

[a] Master repressors of lysogeny are shown in italics, as a reference

[b] Local background is computed as the median of normalized gene expression of all genes of a prophage region (average of the two transcriptomes, S1 Table)

[c] The *sitC* transcription level is indicated (4-fold above background) because it belongs to the *sit* operon.

## The five LF82 prophages are spontaneously induced *in vitro*

We next asked whether some or all of LF82 prophages were producing virions. To identify prophages spontaneously producing viral particles, we filter-purified the virome from the supernatant of an LF82 culture grown in rich medium until stationary phase, treated it with DNase I before destroying capsids, and sequenced the encapsidated DNA. A small part of reads mapped on the *E. coli* LF82 genome out of the prophage regions (4.3% of the reads, mean coverage = 7.6 reads/bp), corresponding to the contaminating bacterial DNA. However, the vast majority of reads (94.6%, mean coverage = 35,319 reads/bp), mapped on Gally (Fig 2),

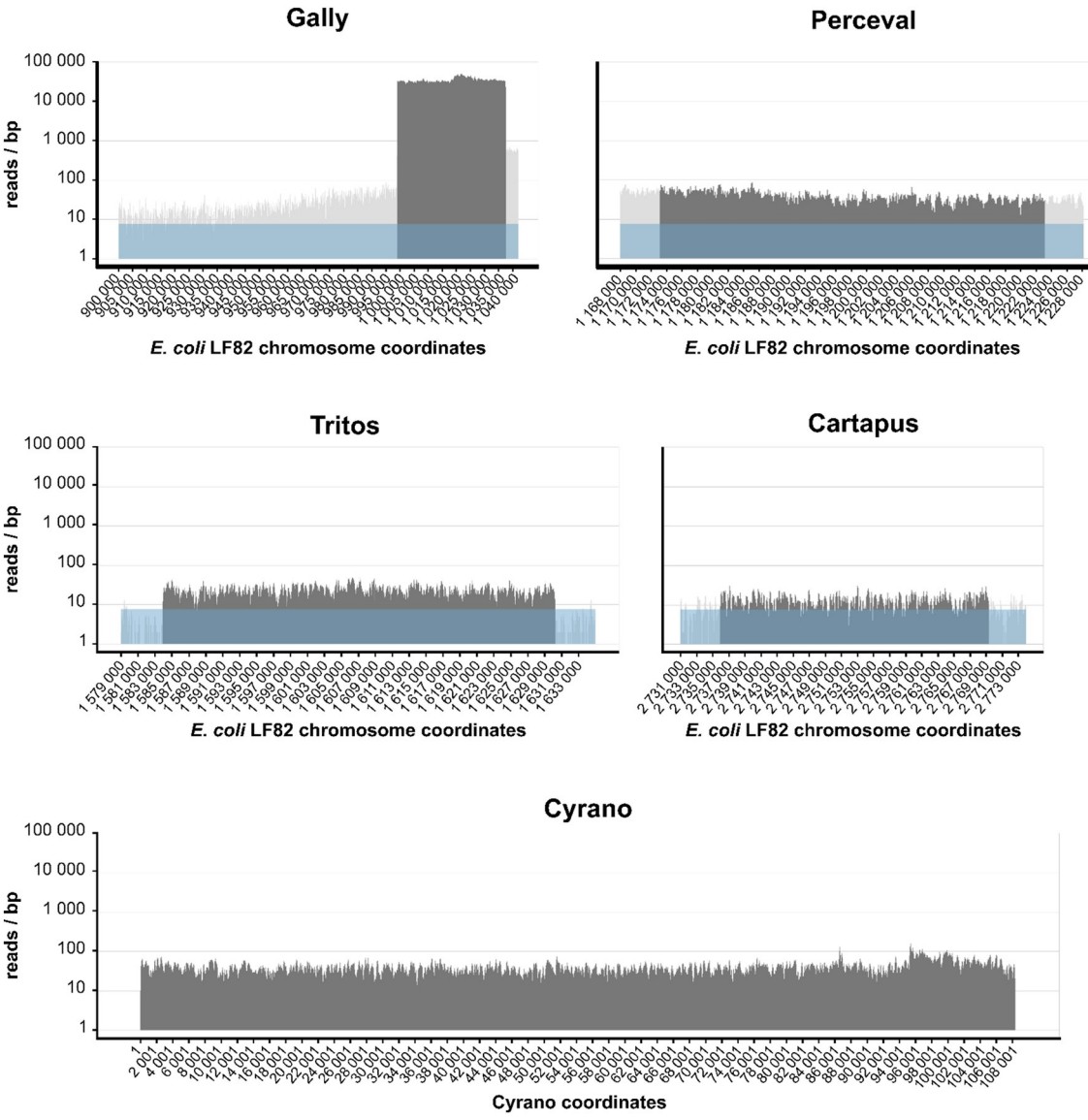

**Fig 2. Shotgun sequencing of the viral particles generated by the five LF82 prophages in an *in vitro* culture in rich medium.**
Nucleotide coverage as a function of LF82 chromosome or Cyrano phage-plasmid coordinates obtained after the mapping of the sequencing reads from the virome of an overnight LF82 culture. The coverages associated with the prophage regions and the bacterial chromosome are shown in dark and light grey respectively. The average coverage linked to bacterial DNA contamination of the virome sequencing is represented in light blue (7.6 reads/bp) for the integrated prophages.

indicating that this phage is highly produced from LF82 cultures (Gally coverage is clearly above the background level, t-test, p-value = $4.3 \times 10^{-8}$). The next most induced prophage was Tritos (0.14% of the mapped reads, mean coverage 24.6, significantly above the background, p-value = $1.6 \times 10^{-11}$) and to a lesser extent Cartapus (0.03% of the mapped reads, mean coverage = 13.7, significantly above the background, p-value = $1.7 \times 10^{-6}$). The activity of Perceval could not be evaluated by this method, due to its proximity with Gally on the LF82 chromosome (~135 kb), combined with the "leakage" of the Gally signal over the Perceval region (see the lateral transduction section below).

The mean coverage of the phage-plasmid Cyrano was 41.2 reads/bp (0.74% of total reads). After determining the copy number of the Cyrano plasmid at 5.5 per LF82 bacteria (S1B Fig),

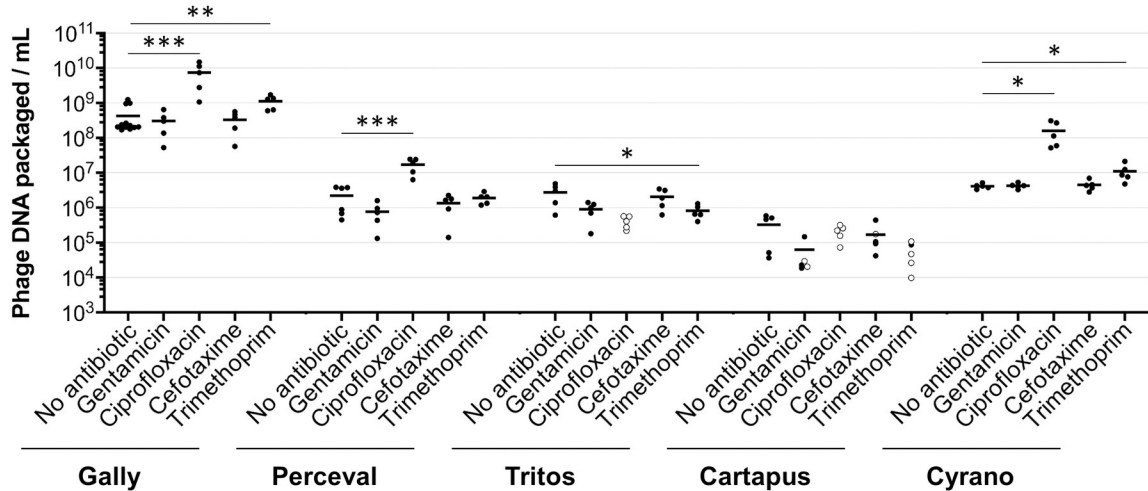

**Fig 3. PCR quantification of viral particles generated by the five LF82 prophages, in rich medium with or without antibiotic, from *in vitro* LF82 culture.** Phages produced from LF82 cells cultured in the presence or absence of an antibiotic, as indicated, were quantified. Each dot corresponds to one biological replicate. Black or white dots correspond to replicates that are respectively more or less abundant than their associated LF82 genomic DNA contamination. Only values corresponding to black dots are used to calculate the mean (vertical line). The statistical difference (t-test) between antibiotic treated and untreated cultures is indicated by one (p-value < 0.05), two (p-value < 0.01) or three (p-value < 0.005) asterisks.

we estimated the contamination associated to the plasmid at 41.8 reads/bp (5.5 x 7.6). The mean coverage of the Cyrano phage was therefore equivalent to the estimated plasmid contamination (p-value = 0.8), and did not allow to conclude whether Cyrano virions were in the LF82 virome.

To confirm the prevalence of Gally in culture supernatants, virion genomes were quantified by quantitative PCR (qPCR), on twelve independent filtered culture supernatants, grown to exponential phase in unstressed conditions. As a control, a qPCR on the *ybtE* gene of LF82, coding for the yersiniabactin biosynthesis salycil-AMP ligase, allowed to estimate a mean value of bacterial contamination for the virome preparations of $2.3 \times 10^4$ chromosomal copies/mL (S2 Table) and $1.3 \times 10^5$ Cyrano prophages/mL ($5.5 \times 2.3 \times 10^4$). Again, a high concentration of $4.3 \times 10^8$ Gally genomes/mL was measured (Fig 3, no antibiotic). Knowing that the bacterial concentration was $6.4 \times 10^8$ CFU/mL, the ratio of Gally virions per bacteria was around 1 in this unstressed growth condition (S2 Table). qPCR were also performed for the other phages and revealed concentrations two to three orders of magnitude below Gally for Tritos and Cartapus, which produced $2.8 \times 10^6$ and $3.3 \times 10^5$ genomes/mL respectively. qPCR highlighted also a production of Cyrano phages at $4.2 \times 10^6$ genomes/mL, two orders of magnitude below Gally, but 32-fold above the contamination associated with the Cyrano prophage ($1.3 \times 10^5$), meaning that Cyrano virions are produced in this condition. qPCR using primers targeting Perceval DNA detected $2.2 \times 10^6$ virions/mL. Due to the lateral transduction of Gally (see below), these genomes correspond either to complete or partial Perceval DNA.

We next attempted to propagate and purify these phages as plaques on an indicator strain, to allow their visualization by transmission electron microscopy (TEM, Fig 4). We succeeded in the isolation and visualization of the two siphoviruses Perceval (capsid diameter (c.d.): 63.3 ± 1.6 nm, tail length (t.l.): 156.8 ± 5.3 nm and tail thickness (t.t.): 11.3 ± 1.5 nm) and Tritos (c.d.: 57.2 ± 9.6 nm, t.l.: 163.5 ± 2.4 nm, t.t.: 11.3 ± 0.4 nm). Gally did not produce any phage plaques under all conditions tested (see Material and Methods). This phage was nevertheless able to lysogenize a LF82 ΔGally host (see Material and Methods), at frequencies $1 \times 10^{-4}$ and

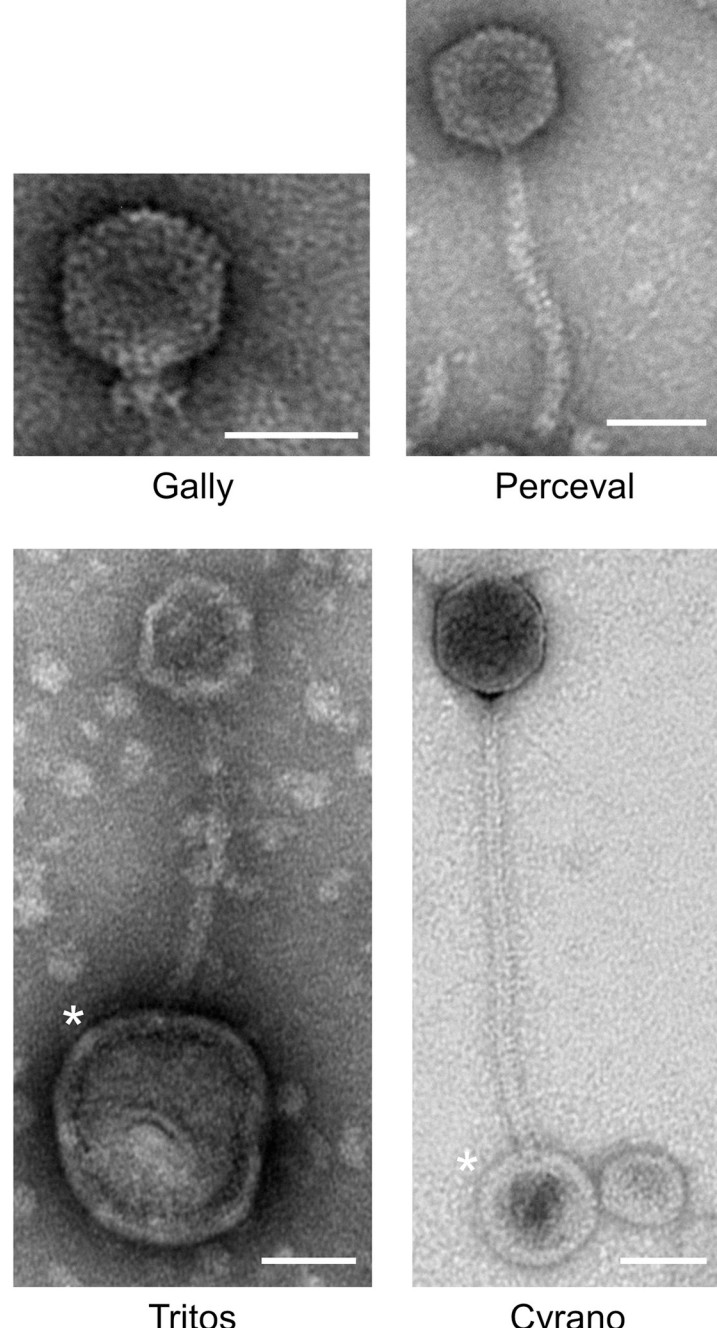

**Fig 4. Transmission electron microscopy photographs of the virions produced by LF82.** Gally and Cyrano were imaged directly from LF82 culture supernatants. Perceval and Tritos phages were visualized after their purification and propagation on MAC1403 as an indicative strain. Asterisks indicate vesicles. Scale bars are 50 μm long.

$3x10^{-4}$ per bacterium, for Multiplicity of Infection (MOI) 1 and 10, respectively. This suggests the phage is infectious, but forms invisible plaques, or 'chooses' lysogeny at high frequency upon infection. Moreover, two Gally-Perceval hybrids (named Galper1 and Galper2, S2 Fig) were isolated in the course of repetitive attempts to isolate Gally plaques on MG1655 *hsdR-* and LF82 ΔGally mutant strains (see Material and Methods). Our study of the LF82 virome

**Table 2. Prophage boundaries on *E. coli* LF82 chromosome.**

| Name | Start | End | Size |
|---|---|---|---|
| Gally | 998,954 | 1,037,635 | 38,682 |
| Perceval | 1,172,883 | 1,223,026 | 50,144 |
| Tritos | 1,583,925 | 1,630,284 | 46,360 |
| Cartapus | 2,735,987 | 2,769,354 | 33,368 |

did not highlight any reads corresponding to the first or the second recombination endpoints of Galper1 and Galper2, meaning that these recombination events are rare (below $4\times10^{-3}$), but lead to the formation of active phages that can be selected and propagated on plate. Galper1 displayed capsid and tail dimensions similar to those of Perceval (c.d.: 62.1 ± 2.3nm, t.l.: 159.1 ± 5.6 nm and t.t.: 12.7 ± 2.1 nm, S2 Fig, panel B), and its genome contained all structural and lysis genes from Perceval, interrupted by the replication module of Gally (S2 Fig, panel A). The junctions consisted in short homology regions (S2 Fig, panels C and D), typical of the substrates used by phage single-strand annealing proteins (SSAP) [31,32]. The rightward recombination junction was identical in Galper2, but the left one (region Rz) was slightly offset (S2 Fig, panels C and D).

In order to visualize Gally virions, we took advantage of its abundance in LF82 culture supernatants and its sequence homology with the HK620 and P22 podoviruses. TEM images of an overnight LF82 culture supernatant showed a huge abundance of a podovirus (c.d.: 69.2 ± 1.9 nm), that we surmised corresponded to Gally (Fig 4). To search for Cyrano virions, we started from a LF82 culture treated with ciprofloxacin (see below) and screened for siphoviruses displaying a tail around 230 nm in length, by applying the 0.15 nm / amino acid ratio between the length of the phage tail and that of its tape measure protein (Cyrano TMP is 1,525 amino acids long) [33,34]. Virions with a large head (c.d.: 76.4 ± 4.0 nm) and a long flexible tail (t.l.: 259.1 ± 11.9 nm, t.t.: 10.3 ± 0.4 nm) were found. This virion being the only siphovirus remaining to identify among phages produced by LF82, and having dimensions typical for SSU5 phages, we hypothesized that it was Cyrano (Fig 4). Finally, no Cartapus-like myoviruses were visualized by TEM, as no recipient strain for its propagation was found, and its abundance was always low in the viromes.

Overall, by combining virome sequencing, qPCR quantification, phage isolation and electronic microscopy observations, we can conclude that under unstressed *in vitro* growth conditions, the five LF82 prophages are induced and form virions (S3 Table). Among them, Gally is by far the most abundant. Of note, for three phages only (Gally, Perceval and Tritos) out of the five, we identified sensitive hosts proving that these phages are also infectious.

Sequencing of the virome as well as the phage attachment sites (*attP*) allowed to characterize the boundaries of the four chromosome-integrated prophages (Table 2, see the Materials and Methods section). *In silico* reconstruction of the bacterial attachment sites and their inspection did not reveal ORFs restoration (>300 bp) after prophage excision, or modification of pre-existing ORFs in the genome of the lysogenized strain, indicating that the integrated prophages do not alter the bacterial gene content after excision. Notably however, Gally was placed between the divergent *torS* (the sensor of the *torS/R* system, pointing leftward) and *torT* (a periplasmic protein pointing rightward). This site is the target of prophage insertion in approximately 5% of *E. coli* strains [35]. The presence of the HK022 prophage at this site increases the expression of *torS* and consequently inhibits the expression of the *torCAD* operon when cells are grown aerobically [35]. The *torCAD* operon codes for a trimethylamine N-oxide (TMAO) reductase that allows the respiration of TMAO by *E. coli* [36].

## Important lateral transduction mediated by the *pac*-type phage Gally

Further analysis of sequencing reads mapping on the LF82 chromosome revealed a particular property of phage Gally (Fig 5A). A mean coverage of the Gally prophage of ~35,000 was observed (dark grey region, Fig 5A), corresponding to the bulk of packaged DNA. This coverage was not homogenous all along the prophage however, as a sharp peak was standing out, localized within the *termS* gene coding for the small terminase subunit (position 1,019,224 on LF82) and showing a gradual rightward decrease until the prophage *attR* attachment site (Fig 5A). Such a pattern is a signature of a headful packaging mechanism, initiated at the *pac* site, localized at the left end side of the peak (Fig 5A).

Furthermore, adjacent to the rightward Gally *attR*, we observed, as mentioned above, a "leakage" where read coverage was well above background, and organized in successive steps of decreasing coverage values. The read coverage immediately downstream the *attR* site was 65-fold higher than the background (coverage ~500 reads/bp compared to 7.6) and was stable over ~20 kb. Following this first reduction, the read coverage decreased about two to three-fold every 40 kb on this side (Fig 5A). Considering that P22 packages ~103.8% of its genome per capsid [37], Gally virions could contain ~40.1 kb of DNA, which corresponds to the approximate length of each decreasing step in the read coverage. This profile of read coverage was described as a consequence of a specific transduction event processed by *pac*-type prophages, called lateral transduction that was initially described for *Staphylococcus* phage 80α [38], and then also reported for P22 [39].

In line with the lateral transduction process, upstream of the *attL* site, the read coverage was approximately 7-fold higher than average DNA contamination (~50 reads/bp *vs* 7.6) and decreased, not by step as observed downstream of the *attR* site, but progressively (Fig 2). This is likely the result of the *in situ* bi-directional replication of prophage Gally upon induction and before its delayed excision.

As mentioned above, Perceval is covered by the lateral transduction area of Gally (Fig 5A), preventing the correct quantification of Perceval virions by DNA-based approaches. Cultures of the LF82 ΔGally mutant in unstressed growth condition led to the production of $1.7 \times 10^5$ particles containing Perceval DNA mL (Fig 5B, S2 Table), confirming that Perceval was an active prophage. Comparison of this amount to that produced by the wild-type strain ($3.7 \times 10^5$ particles containing Perceval DNA mL in this experiment, Fig 5B, S2 Table) indicated that approximately half of the particles containing Perceval DNA were the result of lateral transduction initiated by Gally in the wild-type LF82 strain (t-test, p-value = $6.5 \times 10^{-4}$), assuming a similar induction rate of Perceval in wild-type and ΔGally strains. Perceval is therefore the least produced virion by LF82 in this unstressed condition.

## At least two of the five prophages are induced by ciprofloxacin and trimethoprim

We next investigated whether some antibiotics could induce LF82 prophages beyond their spontaneous level. Genotoxic stresses are known to induce the lytic cycle of many phages *via* the RecA activation and the cleavage of the lysogeny master regulator. We therefore tested antibiotics which activate RecA (and the downstream SOS response) to various extents: (i) ciprofloxacin that inhibits DNA gyrase and topoisomerase IV activities leading to replication fork stalling, (ii) trimethoprim which prevents synthesis of tetrahydrofolate leading subsequently to DNA damages and (iii) cefotaxim, a beta-lactam antibiotics inhibiting primarily the peptidoglycan synthesis but also inducing the SOS response *via* inhibition of the replication [40,41]. We also tested gentamycin, an aminoglycoside that does not induce the SOS response, which we used to eliminate non-invading bacteria upon macrophage infection. The minimal

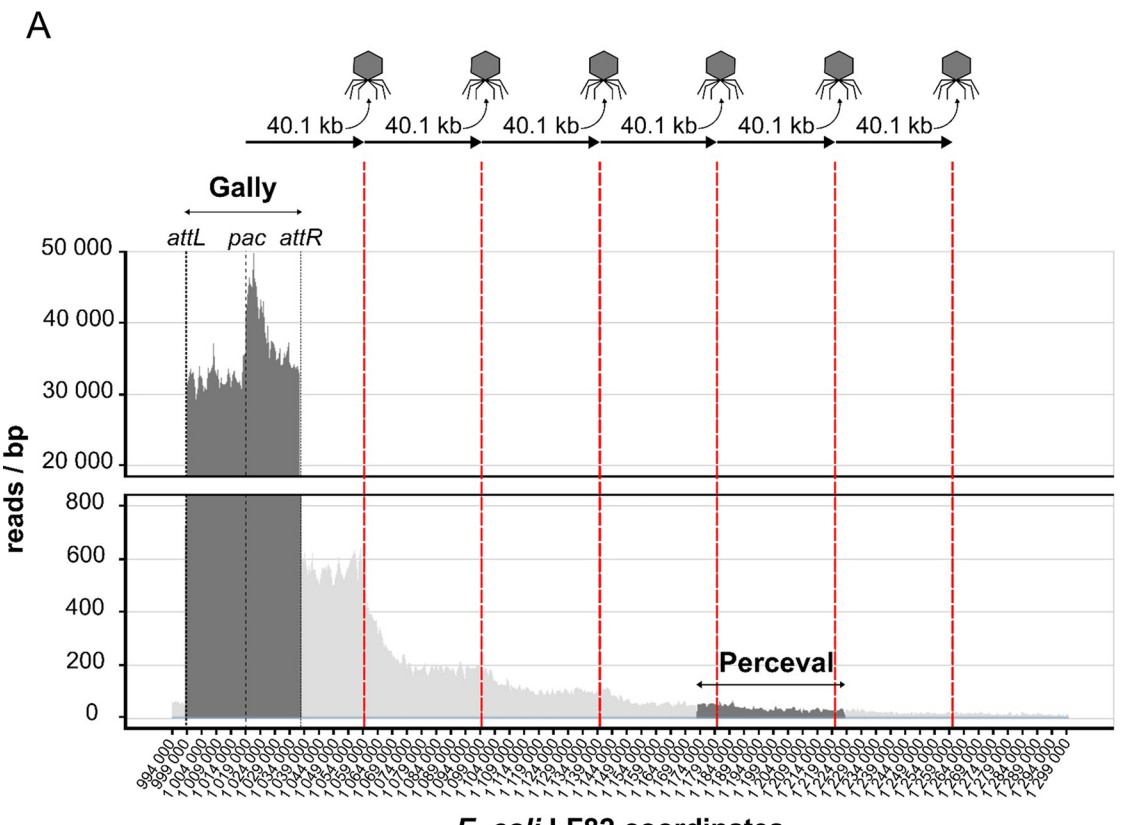

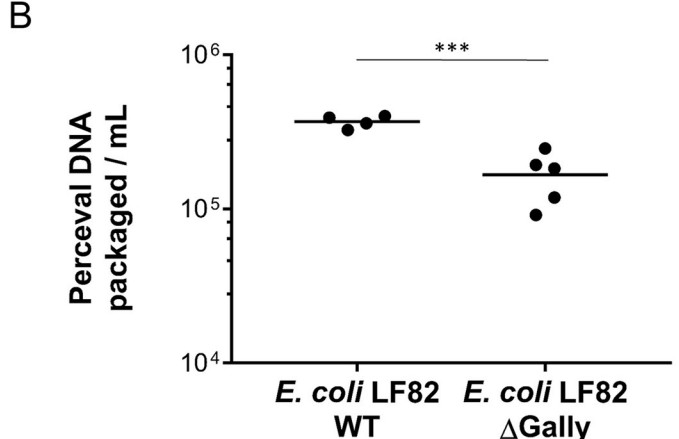

**Fig 5. Gally performs lateral transduction of over 200 kb of adjacent chromosomal DNA, including Perceval.** A. Zoom of mapped reads within the chromosomal region to the right of Gally prophage containing Perceval. The abrupt increase in reads inside Gally corresponds to the *pac* region, from which the packaging is initiated. Red bars indicate 40.1 kb steps of decreasing coverage from the *pac* site of Gally prophage, which could correspond to DNA packaged by a headful mechanism. B. PCR quantification of Perceval virions produced from an *in vitro* culture in rich medium of a wild-type or a Gally-deleted strain of *E. coli* LF82. Each black dot corresponds to one biological replicate and black horizontal lines indicate the mean values for each condition. The statistical difference (t-test) between the two strains is indicated by three asterisks (p-value = 0.0006).

inhibitory concentrations (MICs) for these antibiotics were first determined in *E. coli* LF82 (see Material and Methods). Then, antibiotics were added at concentrations corresponding to the MIC in exponentially growing cultures, and supernatants were harvested two hours later, filtered and virions were quantified by qPCR. Gally and Cyrano were strongly induced by ciprofloxacin, 18 and 39-fold respectively (Fig 3). We cannot conclude whether ciprofloxacin also induced Perceval, as the 8-fold increase in Perceval copy number was probably a consequence of the lateral transfer activity of Gally. Trimethoprim had a mild induction effect on Gally and Cyrano (3-fold), and decreased slightly (3.4-fold) the production of Tritos. Finally, cefotaxime and gentamycin did not have any effect on the induction level of the five prophages. We conclude that a genotoxic stress similar to the one provoked by a 2 hours ciprofloxacin-exposure at the MIC strongly induces part of the LF82 phageome.

## LF82 survival in macrophages is not affected by the presence of the Gally prophage

Gally is the most produced virion after ciprofloxacin treatment. This suggests that Gally is the main cause of LF82 lethality after genotoxic stress. We tested this hypothesis by comparing the survival of wild-type and ΔGally strains after exposure to ciprofloxacin at the MIC in rich medium. A typical phage induction-dependent cell lysis was observed for wild-type bacteria between 50 and 80 minutes after ciprofloxacin addition (Fig 6A). In contrast, during this period, the $OD_{610}$ of the ΔGally culture remained stable. After 80 minutes, both cultures behaved similarly with a slight and steady decrease in OD over time. Thus, the lysis of wild-type bacteria after ciprofloxacin-mediated genotoxic stress is indeed primarily due to Gally induction. This *in vitro* observation raised the question of a putative negative effect of Gally on the survival of LF82 inside macrophages, an environment that provokes genotoxic stress to LF82 highlighted by the SOS response level [25,27]. To test this hypothesis, we compared the survival of the two strains upon macrophage infection (Fig 6B and 6C): survival was not significantly increased with the mutant compared to the wild-type strain, neither at 6 hours nor at 24 hours post-infection, showing that the macrophage survival of LF82 is not diminished by the presence of the Gally prophage.

## Gally transcription is partial in macrophages

The absence of any negative effect of Gally on the survival of LF82 in macrophages strongly suggested that Gally particles were not produced in this genotoxic environment, in contrast to the *in vitro* ciprofloxacin treatment. We therefore investigated more precisely the fate of Gally upon macrophage infection.

Using our previously published transcriptomic analysis [27], the transcription profiles of the Gally prophage in LF82 bacteria internalized in macrophages (6 hours post-infection) or cultured *in vitro* in LB to stationary phase were compared (Fig 7, S4 Table). The LB profile revealed high levels of the C2 repressor transcript (functional homolog of the Lambda CI). Within macrophages, the Gally region exhibited a clearly different pattern: first, the five rightmost genes of the prophage, including the genes coding for the putative Mnt repressor and a tail spike, were highly induced (6 to 40-fold, S4 Table). In addition, on 13 genes in the leftward region of the prophage (from gene *c2* down to the last gene before *xis*), 9 were statistically overexpressed (2 to 11-fold above their *in vitro* level), in particular the recombination module (*recT* gene, 3.6-fold, q-value = $3x10^{-5}$). In the leftward region up to the antiterminator *Q* gene, some genes were upregulated (replication initiation *O* gene, 2-fold, q-value = $7.6x10^{-3}$), while downstream of *Q*, several key structural genes, including *terL* and *portal* as well as two genes coding for DNA injection proteins, were statistically repressed 2 to 7-fold (Fig 7, S4 Table).

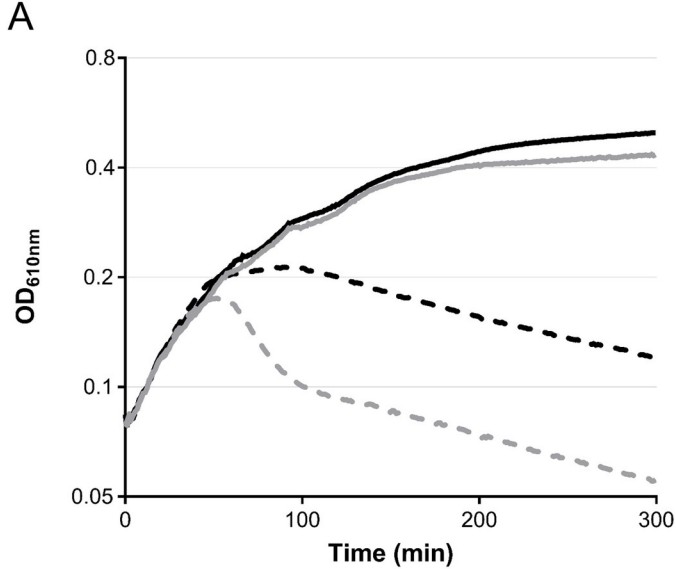

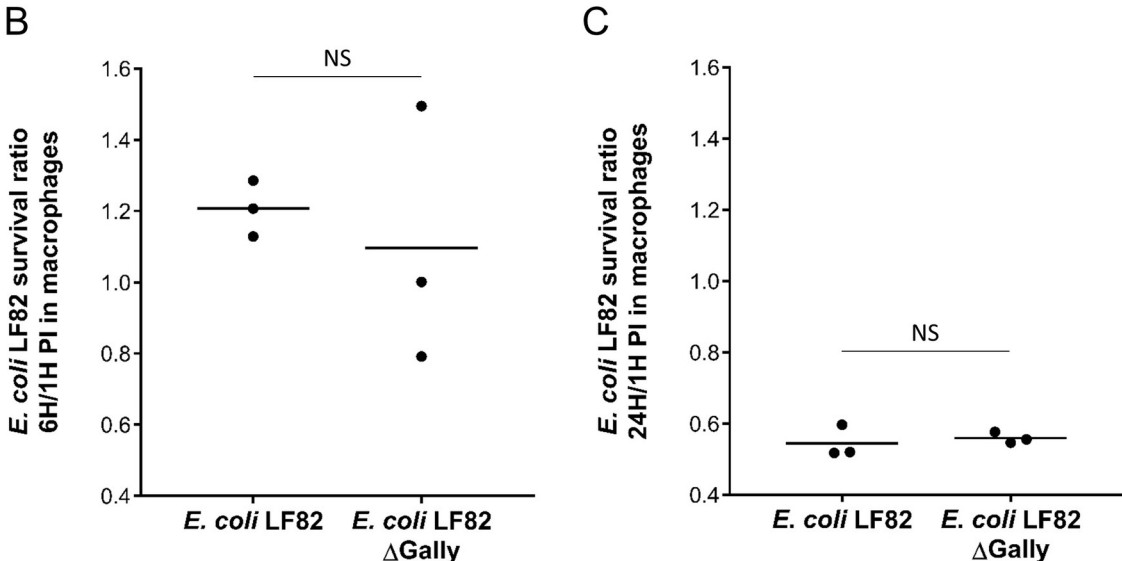

**Fig 6. The survival of LF82 treated *in vitro* with ciprofloxacin is affected in the presence of the Gally prophage, contrary to what is observed after THP-1 macrophages infection.** A. Optical density of *in vitro* cultures of wild-type (grey lines) or ΔGally (black lines) LF82 bacteria was monitored after addition (dashed lines) or not (continuous lines) of ciprofloxacin at the MIC. Experiments were performed three times and quantifications shown are from one representative experiment. B. and C. Survival (CFU/mL) of the wild-type or ΔGally strains, after 6 (B) or 24 (C) hours in THP-1 macrophage was compared to the initial amount of endocytosed LF82 bacteria (CFU/mL at 1-hour P.I. as reference). Black dots represent values from three biological replicates obtained after independent macrophage infections. Horizontal black lines represent mean values. NS: not significant (t-test, p-values > 0.6).

Finally, antisense transcripts were also observed specifically in macrophages, especially in the region starting at *Q*, and covering up to the *roi* gene (Fig 7). This suggests that a macrophage-dependent regulation takes place at the level of *Q*.

Among the five transcriptional regulators encoded by Gally, the *mnt* repressor gene was upregulated 40-fold (q-value = $2.5 \times 10^{-17}$) in macrophages, *c2* and *c1* were upregulated 3 and

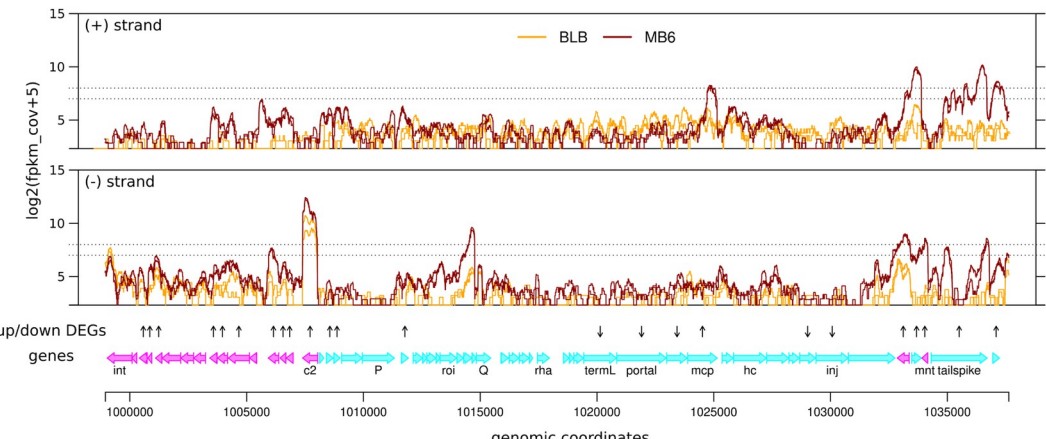

**Fig 7. Transcription of the Gally prophage in macrophages.** Comparison of coverage by RNA-Seq fragments along Gally genome between THP1 macrophages infected by LF82 at 6h P.I. (MB6, dark red) and LF82 bacteria grown in LB to stationary phase (BLB, orange). Data from two biological replicates are shown for each condition along the region corresponding to the prophage (from 998,954 to 1,037,635 bp) in *E. coli* LF82 genome. From top to bottom: transcription profiles on + and – strands, expressed in log2(fpkm+5); vertical arrows indicating genes detected as differentially expressed (q-value ≤ 0.01 and |log2FC| ≥ 1), pointing upward for up-regulated in macrophages, downward for down-regulated; genome annotation (names for selected genes). The vertical distance between horizontal dotted lines in the transcription profile panels correspond to a log2FC of 1.

16-fold respectively, and *roi* and *rha* were unaffected. Since the *mnt* repressor gene was the most highly and specifically overexpressed within Gally in macrophage, we tested whether it was involved in the repression of the Gally lytic cycle in this environment. The Δ*mnt* strain survived as well as the LF82 wild-type strain after 1, 6 or 24 hours post-infection in macrophages (t-test, p-values>0.5, S3 Fig), suggesting that Mnt is not involved in this regulation of the phage cycle.

## Gally induction is blocked in macrophages

To further explore the possibility of a partial repression of the Gally lytic cycle in macrophages, we attempted to quantify Gally particles in this environment. First, a spike-in experiment of a macrophage lysate with Gally particles at high concentration ($10^8$/mL) showed that they were rapidly degraded (20-fold decrease upon 12 hours incubation at 4˚C). Next, a 6 hours LF82 infection experiment was conducted as usual, and macrophage were lysed. Part of the lysate was used directly for bacterial counts (S2 Table), and the rest was kept 12 hours at 4˚C before qPCR processing. Taking into account the instability factor and the bacterial DNA contamination of the samples, Gally was quantified at $1.4 \times 10^6$ particles/mL of macrophage lysate (S2 Table). Under these conditions, a phage/bacterium ratio of $6.3 \times 10^{-1}$ was estimated, which is similar to the *in vitro* ratio for an unstressed culture ($7.2 \times 10^{-1}$), but 65-fold less than the ciprofloxacin treated cultures (Fig 8A).

Cyrano is also induced in the presence of genotoxic stress, its behavior in macrophages was therefore investigated. Cyrano particles were not detected above background levels (S2 Table). Given that these phage particles are not detectable after macrophage infection, we estimate that the maximum phages per bacteria ratio for Cyrano in macrophages is also equivalent to the one obtained in an *in vitro* Lennox condition (Fig 8A), showing that, as for Gally, the Cyrano lytic cycle seems to be repressed in macrophages.

To determine whether a change in Gally induction frequency contributes to these net shifts, we constructed a LF82 strain in which the gene coding the major capsid protein (MCP) of

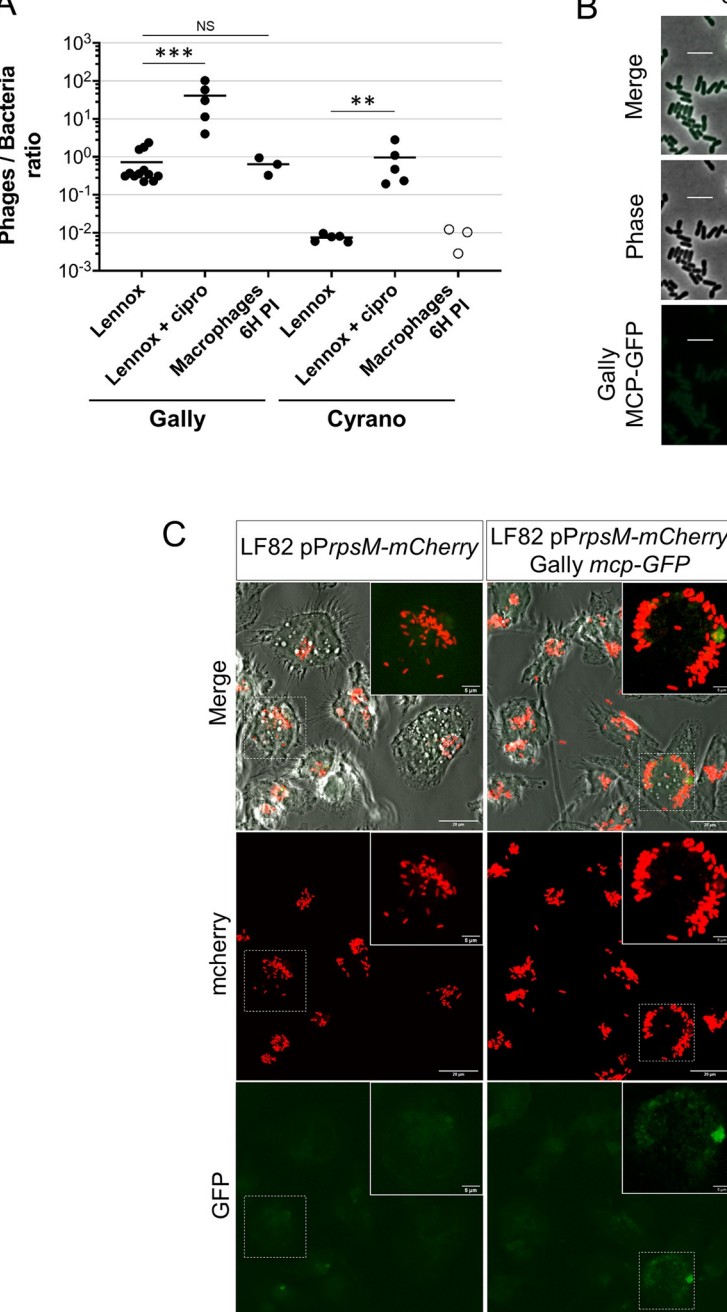

**Fig 8. Gally and Cyrano are not induced in LF82 upon macrophage infection.** A. Comparison of phages/bacteria ratios obtained for Gally and Cyrano, after cultures *in vitro* in Lennox and Lennox + ciprofloxacin (MIC), and at 6 hours P.I. in macrophages. Black or white dots correspond to replicates that are respectively more or less abundant in phages than their associated LF82 genomic DNA contamination. Only values corresponding to black dots are used to calculate the means (vertical lines). Statistical differences (Mann-Whitney test) between ciprofloxacin-treated (Lennox +cip) or macrophage infection (Macrophages 6H P.I.), and untreated cultures (Lennox) is indicated by two (p-value < 0.01) or three (p-value < 0.005) asterisks. B. Gally phage induction was followed by the MCP-GFP fusion production in *in vitro* ciprofloxacin-treated LF82 cultures. Snapshots of LF82 bacteria (strain MAC2606) 60 minutes after ciprofloxacin treatment (+cip) or of untreated cells (-cip) are shown. Scale bars correspond to 5 μm. C. Confocal imaging of THP-1 macrophages at 6 hours P.I. with the LF82-pP*rpsm-mCherry* (OEC2425) (left panel) or LF82-pP*rpsm-mCherry* Gally *mcp-GFP* (OEC2481) (right panel). White framed areas correspond to zooms of the white dashed framed parts. Scale bars indicate 5 or 20 μm, as indicated.

Gally is fused to the GFP. To validate the use of this fusion as an induction marker, we followed the fate of LF82 *Gally-mcp-GFP* (MAC2606 strain) plated on minimal medium supplemented with ciprofloxacin at the MIC in agar pads deposited on slides (S4 Fig). Four categories of cells were identified: (1) non-fluorescent cells that were intact or (2) lysed over time, (3) MCP-GFP-expressing cells, identified as homogeneously fluorescent cells or cells with fluorescent foci, intact or (4) lysed. Intact MCP-GFP-expressing cells were observed as early as 60 minutes of exposure, with a maximum amount (16% of the 189 cells analyzed) after 90 minutes. Fluorescent cells began to lyse approximately 20 min later (S4 Fig panel B). After 160 min, 36% of the observed cells have expressed the MCP-GFP fusion during incubation, and 96% of them have lysed (S4 Fig panel C). Cells not expressing MCP-GFP also lysed over time, but later than fluorescent cells and in a lower proportion (31.5%). We conclude that the induction of the Gally MCP-GFP in the presence of ciprofloxacin is a good marker for the induction of its lytic cycle, even if this fusion did not allow the formation of Gally virions (as verified by qPCR).

We next estimated the induction frequency of Gally in rich medium (Fig 8B). A population of 1-hour ciprofloxacin treated cells, *i.e.* the time point just before the massive ciprofloxacin-dependent lysis observed in Fig 6A, produced 44% of intact or recently lysed bacteria with fluorescent dots (616 fluorescently labelled cells on 1,387 observed). In the absence of drug, the frequency of spontaneous Gally induction was 0.37% (20 fluorescently labelled cells on 5,337 observed, Fig 8B).

Finally, Gally induction in macrophages was investigated. None of the imaged LF82 *mcp-GFP* bacteria displayed fluorescent dots, and only one lysis event was detected over a total of 2,232 observed bacteria (Fig 8C, 6 hours post-infection, S5 and S6 Figs for 40 min and 24 hours post-infection respectively). We conclude that Gally induction frequency is approximately 0.04% in LF82 infecting macrophages, around 10-fold lower than *in vitro* unstressed growth conditions.

## The Gally lytic cycle is blocked at the excision/replication stage in macrophages

To investigate more precisely at which stage of its lytic cycle Gally was blocked upon growth in macrophages, we sought to quantify its excision and replication. For this, primers were designed to detect the Gally *attL* (marker for bi-directional *in situ* replication of the prophage within the bacterial chromosome), *attB* (formed after excision of the prophage from the bacterial chromosome) and *attP* (reconstituted after circularization of the phage-excised genome, and amplified by the replication of the circularized form of the genome) sites. Quantity of PCR products obtained were compared to that of the bacterial *ybtE* gene used as reference. The DNA templates used were extracted from (i) macrophages infected with LF82 bacteria during 6 hours, (ii) *in vitro* cultures in Lennox (control of Gally spontaneous induction) and (iii) *in vitro* cultures treated for 1 hour with MIC of ciprofloxacin (DNA extracted just before the Gally-dependent lysis, control of Gally induction, Fig 6A). The sequences surrounding the *attL*, *attB*, and *attP* sites did not allow us to design primers suitable for qPCR analysis, so we performed semi-quantitative analysis by standard PCR. As expected, *in situ* replication (*attL*), excision (*attB*), and replication post-circularization (*attP*) of Gally were more frequent in LF82 cultures treated with ciprofloxacin than in untreated cultures (S7 Fig). Interestingly, bacterial DNA extracted from macrophages showed lower levels of excision and replication post-circularization than those quantified in untreated cultures. In contrast, *in situ* replication appeared to take place in macrophage, as the *attL/ybtE* ratio was equivalent to the one observed in unstressed *in vitro* growth conditions. These results strongly suggest that the Gally lytic cycle is repressed at the excision stage upon macrophage infection.

## Discussion

The five predicted prophages of strain LF82 were spontaneously produced in exponential growth phase under unstressed culture conditions. However, based on our quantification of free virions (unadsorbed at the bacterial surface), a clear gradation was observed: Gally, a P22-like podophage had the highest level of virions production (ratio of ~1 virion/bacterium), Cyrano and Tritos, a SSU5-like and a Lambda-like phage respectively, produced some 100 to 150-fold less free virions than Gally. Finally, spontaneous production levels were the lowest (1,300 to 2,500-fold lower than Gally) for Cartapus and Perceval (estimated from the ΔGally mutant for the latter), a P2 and a Lambda-like phage respectively.

The high abundance of Gally virions in the supernatant of LF82 bacteria grown exponentially is unusual but not so exceptional. For instance, prophage BTP1 from *Salmonella typhimurium* ST313, that shares homology with P22 and HK620, is also highly spontaneously induced *in vitro*, giving rise to $10^9$ virions/mL of a stationary phase culture of its host strain [42]. This high production of virions is however not a hallmark of P22-like viruses since P22 itself is ~10,000-fold less produced than BTP1 *Salmonella enterica* serovar Typhimurium LT2 [42]. Our estimation of the spontaneous induction of Gally *in vitro* (0.37%) is also similar to the one calculated for BTP1 (~0.2%, [42]). As Gally, BTP1 is induced by a genotoxic agent that promotes SOS response [42]. However, unlike BTP1 [42], Gally virions were not able to form visible plaques on their Gally-deleted host in our test conditions, but they could lysogenize it.

A transcriptome analysis allowed to uncover systematically the morons encoded by these prophages. We demonstrate the presence of 29 moron genes, with nearly half of them being of unknown or poorly characterized function. Clearly, efforts should be placed in the future to better understand the biological function of phage moron genes in natural environments. Interestingly, the Gally prophage may be frequently present in *E. coli* strains associated to Crohn disease patients [23], but it does not carry any moron genes in its genome.

The elevated production of Gally virions allowed to highlight its lateral transduction activity. Lateral transduction was first described for prophages of *Staphylococcus aureus* [38], and then identified as well for the phage P22 [43] and phages from *Enterococcus faecalis* VE14089 [44]. Whether this lateral transduction contributes to the expansion of *E. coli* strains adapted to survival in dysbiotic microbiota is unknown at present. We searched for virulence or adaptation genes in the transduced regions, and found none, except those present within the Perceval prophage. Indeed, we estimated that half of particles containing Perceval DNA were probably Gally transducing particles. Prophage evolution might therefore depend in part on this lateral transduction process, whereby a region of a Lambda-like prophage could be exchanged for Gally-transduced Perceval genes. Perceval encodes an operon of morons with a function relevant for different human environments, the SitABCD transporter. It might help LF82 bacteria to scavenge metal ions (iron or manganese) during macrophage infection. We also noted that LF82 grown on the DMEM medium used for macrophage propagation induced expression of additional Perceval morons such as the EmrE multidrug exporter and the Bor lipoprotein, which might be beneficial as well for the intra-vacuolar life-style of LF82. But what Gally's lateral transduction actually brings to its LF82 host remains unknown at this time.

Bodner *et al*. demonstrated recently that a K12 Lambda lysogen has some 30-fold increased levels of prophage induction inside macrophages compared to the *in vitro* induction observed on an agar pad [16]. We show here that this is not the case for the AIEC *E. coli* LF82, at least for its most active prophage *in vitro*, Gally, whose particles are barely detected in macrophages, whereas they abound *in vitro* in the presence of ciprofloxacin. Interestingly, Lambda induction inside macrophages was reported to depend on *phoP* gene expression [16]. Even if PhoP was a general prophage induction pathway in macrophages, the *phoP* gene is transcribed in

phagocytosed LF82 bacteria (see S4 Table), excluding that the Gally defect could be due to a lack of PhoP. It is more likely due to the marked repression of its lytic cycle in macrophages, since it is around 10-fold less induced than in unstressed *in vitro* conditions, as estimated by quantification of cells expressing MCP-GFP. At what stage exactly this cycle is blocked is unknown at present. Transcriptomic data suggest that a dedicated control prevented the transcription of genes needed for phage DNA packaging. We also noticed that genes encoding Gally integrase and excisionase were not overexpressed in macrophages, unlike many of the genes to the left of *c2* (S4 Table). This latter observation could explain the decrease in excision events and replication of the circular form of the phage in macrophages compared to unstressed growth *in vitro*. In contrast, the *in situ* replication of the prophage in the bacterial chromosome is quite similar in both conditions. Moreover, the overexpression of *c1* (homologous to Lambda *cII*) in macrophages strongly suggests its implication in the "super lysogeny" of Gally in this environment, possibly by activating the Q antisense transcript (Fig 7).

Our study reveals that LF82 has evolved in order to control the lytic cycle of Gally inside macrophage, a prophage that is highly produced in other genotoxic growth conditions, rather than deleting it. Despite the fact that Gally prophage is apparently conserved in *E. coli* genomes associated to Crohn's disease [23], its deletion did not affect the survival of LF82 inside macrophages, indicating that Gally has no role in this cellular environment. This is in contrast to the φ10403S prophage of *Listeria monocytogenes* 10403S, which performs active lysogeny. This mechanism blocks the lytic cycle of the phage after the excision of its genome and is required for the survival and multiplication of bacteria within macrophages [45]. However, it is possible that Gally provides some benefits to its host in other environments. Indeed, the Gally prophage is inserted between the *torT* and *torS* genes involved in the regulation of the *torCAD* operon coding for TMAO reductase [36]. Prophage integration at this site is known to affect the regulation of *torS* and consequently of the *torCAD* operon [35]. We found a putative promoter in Gally, positioned at a place similar to the one characterized at the left boundary of the prophage HK022, which regulates the expression of *torS* in *E. coli* MG1655 [35]. This suggests that Gally, as HK022, modulates the expression of the *torCAD* operon. However, the transcription profile of the Gally prophage neighboring genes, *torS* and *torT*, was not affected upon macrophage infection with LF82 (S4 Table), in line with the observed low excision of Gally prophage in this environment. The lack of a role in macrophages does not exclude a role of Gally in a different setting, such as in the gut lumen or in external environments where TMAO is also available. TMAO respiration is performed by *E. coli* under anaerobic and aerobic conditions [46]. In the latter case, only part of the bacterial population expresses *torCAD*, leading to the proposal that TMAO respiration under aerobic conditions may facilitate bacterial adaptation to anaerobic conditions [47]. The insertion of HK022, and most likely Gally in LF82, between *torT* and *torS* represses *torCAD* expression under aerobic conditions, and thus should inhibit this adaptive advantage [35]. In this latter study, the authors proposed that the phage-dependent repression of *torCAD* could increase the growth rate of the bacterial host in the presence of oxygen and TMAO, and thus the dissemination of the phage, most probably outside the gut. Remarkably, the HK022 prophage does not inhibit *torCAD* induction under anaerobic growth conditions [35], indicating that phage insertion at this site should not affect the physiology and competitiveness of a lysogenic strain in the gut. Thus, Gally could increase, when not induced, the competitiveness of LF82 under certain aerobic growth conditions. The subpopulation sacrificed by the prophage induction would allow the dispersion of LF82 genes by lateral transduction to recipient strains, yet to be identified. Clearly, the intricate details of the "symbiosis" between a temperate phage and its host have novel shades that we just start uncovering.

## Material and methods

### Bacterial strains

Table 3 lists the bacterial strains, and Table 4 the oligonucleotides used in this study. Unless otherwise stated, cultures were propagated in LB Lennox broth (5 g/L NaCl instead of 10 in regular LB), at 37°C under agitation.

The LF82 ΔGally strain (MAC2225) was obtained by curation of the prophage using ciprofloxacin. For this, a culture of exponentially growing LF82 in LB at 37°C ($OD_{600}$ = 0.2) was diluted ten-fold and treated with 2 μg/mL ciprofloxacin during 30 minutes. Cells were washed with LB and plated on LB agar plates. After incubation at 37°C, individual colonies were screened by PCR for the absence of the Gally prophage using primers JC206 and JC207 that hybridize downstream and upstream the prophage on the bacterial chromosome. Integrity of the *attB* region remaining upon excision was verified by sequencing the PCR fragment generated.

Strain MAC2459 (Δ*bla Gally-mcp-GFP*, KanR) was obtained by integration in MAC2218 of a mEGFP-KanR cassette at the 3' end of the *mcp* gene of the Gally prophage, by recombineering [48]. In a prior step, to place *mEGFP* next to a kanamycin resistant gene, *mEGFP* was amplified with OFL311 and OFL312 from pKD3 pR:*mEGFP* (I. Matic, plasmid collection) and cloned into a ClaI/BmgBI double digested pKD4 [48]. The resulting plasmid, pFL111, was then used to generate the PCR substrate for MAC2459 construction, with oligonucleotides OFL321 and OFL322. The KanR cassette was then deleted from MAC2459 *via* a Flp-FRT recombination, using the pCP20 plasmid [49], giving strain MAC2606.

Strain MAC2774 (LF82 Δ*bla* Δ*mnt*::kanR) or MAC2222 (LF82 Δ*recT-gam-abc1*::kanR) were obtained by integration in MAC2218 or MAC2204, *via* recombineering, of the KanR cassette of the pKD4 plasmid amplified with OFL383/OFL384 or OPM5/OPM6 respectively.

Strain OEC2481 was obtained by transformation of the MAC2606 with plasmid pP*rpsm-mCherry* that expresses mCherry constitutively [27].

### Prophage detection and annotation

Prophage region predictions on the *E. coli* LF82 genome (NC_011993, [22]) were initiated with PHASTER (http://phaster.ca [52]). Regions containing phage genes from replication,

**Table 3. Strains used in this study.**

| Strain | Genetic background | Origin |
|---|---|---|
| MAC2204 | *E. coli* LF82 | NC_011993, [22] |
| MAC2218 | *E. coli* LF82 Δ*bla* | [26] |
| MAC2225 | *E. coli* LF82 ΔGally | This study |
| MAC2459 | *E. coli* LF82 Δ*bla Gally-mcp-GFP*, KanR | This study |
| MAC2222 | *E. coli* LF82 *Gally* Δ*recT-gam-abc1*::KanR | This study |
| MAC2606 | *E. coli* LF82 Δ*bla Gally-mcp-GFP* | This study |
| OEC2481 | *E. coli* LF82 Δ*bla Gally-mcp-GFP*, pP*rpsm-mCherry* | This study |
| OEC2425 | *E. coli* LF82 Δ*bla* pP*rpsm-mCherry* | [27] |
| MAC1403 | *E. coli* MG1655 *hsdr*::KanR | [2] |
| MAC2266 | *E. coli* C+ | ATCC8739 |
| MAC2294 | *E. coli* TD2158-PL4 (phage-free, susceptible to HK620 phage) | [50] |
| MAC2310 | *E. coli* UGB 2668 (MG1655 *wbbl*+) | Pasteur Institute |
| MAC2267 | *E. coli* DH10B | [51] |
| MAC2774 | *E. coli* LF82 Δ*bla Gally* Δ*mnt*::KanR | This study |

**Table 4. Oligonucleotides used in this study.**

| Oligo-nucleotide | Region targeted | Sequence | Use |
|---|---|---|---|
| OPM7 | Gally gene downstream *recT* | AACCGAAAGAGTTAAGGCTG | LF82 phage identification by PCR |
| OPM8 | region after Gally *abc1* | GCGAATAATCATTTCTGCCG | |
| OPM15 | Perceval *xis* | TCAGCGATTATCCGTTGGAG | |
| OPM16 | region after Perceval *xis* | CCTGCCATGAGCTTAATATC | |
| OFL290 | Perceval *tail* | AGCGGCAGTCGTTGAACAG | |
| OFL291 | Perceval *tail* | GGCTGATGGACGCAATCTG | |
| LF-ph3-port1 | Tritos *portal* | ACTGCTGCCTCCTTTATCAC | |
| LF-ph3-port2 | Tritos *portal* | GGAACTGGTTGAGTCTACTG | |
| LF-ph4-con1 | Cartapus *portal* | GGGTTAGACCTGGATACCTAC | |
| LF-ph4-con2 | Cartapus *portal* | AAGTTGCCTGGACCTTTGG | |
| JC146 | Cyrano *int* | CTGGGCTATTGCCACTTTAGACATC | |
| JC147 | Cyrano *int* | CCAACTGGTCAACCCACTAATACTG | |
| OPM50 | 1st recombination endpoint | AGGGAGAACCTGTCTGTATC | Comparison of recombination junctions between Galper1 and Galper2 |
| OPM51 | 1st recombination endpoint | TGGCAATCCAGTGCAAAG | |
| OPM52 | 2nd recombination endpoint | ATGGATCGCGGCTATTTC | |
| OPM53 | 2nd recombination endpoint | TTCACGCCTCAATAACCC | |
| OPM75 | Gally *attL* site | AAACCTTGTTCCCGTAACGC | |
| Maj281 | Gally *attL* and *attP* sites | CGTCTTCTCGGGCATAAATC | |
| Maj280 | Gally *attP* site | TTGCGCTAATGCTCTGTC | |
| JC206 | Gally *attB* site | GCGCCATATTCATGGTAG | |
| JC207 | Gally *attB* site | TTAAGCGGCGTAGAGGCTTG | |
| OPM80 | LF82 *ybtE* | GGCTCAGCGCGTGGAA | quantification of phages and sites by PCR or qPCR |
| OPM81 | LF82 *ybtE* | CGGCCAGTGGTCCAGAAA | |
| OPM82 | LF82 *ybtE* | GACGCCATCGACATACAGG | |
| JC78 | Gally *injection* | GCCTTGCGTCATCTTCTCCA | |
| JC79 | Gally *injection* | TCTGAGCAACGCTGTTAGGG | |
| OPM46 | Perceval *ycbk* | GCATGGGGGCCTTCTGTAA | |
| OPM47 | Perceval *ycbk* | GCCAGCGATTTCACTTATCCC | |
| JC88 | Tritos *minor tail* | CATCCCGGTGACCATGCC | |
| JC89 | Tritos *minor tail* | ACGGGATTTGAACTGAACGGTA | |
| JC100 | Cartapus *tail* | TTGTCCAGCGGTTGTTTACCT | |
| JC101 | Cartapus *tail* | CGGCACTGGATACACTGAAC | |
| JC96 | Cyrano *repA* | TGACAAGTCGCACTATTACTCAGAA | |
| JC97 | Cyrano *repA* | CTCGCAGCTGTTCCATAGCC | |
| OFL311 | *mEGFP* | AACCATCGATATGGTTAGCAAGGGCGAGG | Construction of pFL111 |
| OFL312 | *mEGFP* | TCACTTGTACAGCTCGTCC | |
| OFL321 | Tagging of Gally *mcp* with a mEGFP-kanR cassette | TATGTGTGCTTTAACCCTCACATGGGCG GTCAGTTCTTCGGTAATCCGATGGTTAG CAAGGGCGAG | Induction of Gally *via* MCP-GFP production |
| OFL322 | Tagging of Gally *mcp* with a mEGFP-kanR cassette | AACTTACGAAGCGCAAAAAGGACGATC TCACCCTTTGTCAGTACTGTTGCCATATG AATATCCTCCT | |
| OFL383 | Amplification of the kanR cassette on pKD4 | ATACCATCAACAAAGCAAGACTAATAAAT AGGACCCACCTGTGTAGGCTGGAGCTGCTTC | Deletion of *mnt* of Gally (strain MAC2274) |
| OFL384 | Amplification of the kanR cassette on pKD4 | AATAAGATGCCGATCCACTCACAAAAGCGA GGCATCAAGAATGAATATCCTCCTTAGTTC | |

*(Continued)*

**Table 4.** (Continued)

| Oligo-nucleotide | Region targeted | Sequence | Use |
|---|---|---|---|
| OPM5 | Amplification of the kanR cassette on pKD4 | ATGCCTTCGCAATATTCAAACGCAGACACAT TTTTGGAGAAGCAGCATGAGTGTAGGCTGG AGCTGCTTC | Deletion of *recT-gam-abc1* of Gally (strain MAC2222) |
| OPM6 | Amplification of the kanR cassette on pKD4 | GAAGAATGCCGGGATTGTATGCAAGTCCTCT CATGGTAAATTCCTCTTTGCATATGAATATCC TCCTTAG | |

capsid, lysis and lysogeny modules were confirmed as complete prophages; we further verified the absence of genes specific of integrative plasmids or insertion sequences. To annotate hypothetical genes, a BLASTP search against the viruses taxid 10239 database was performed with default values, and the annotation of sequences producing significant alignments were transferred to the query when either experimental evidence of function or conserved domains were detected. For hypothetical proteins without BLASTP hit, sequences were analyzed for Pfam matches [53].

## Transcriptome analyses

RNA-Seq raw data from GEO accession GSE154648 (10 samples with bacteria from 5 conditions) were reanalyzed (S4 Table). Reads were mapped onto *E. coli* LF82 and Cyrano genomes (Genbank accession numbers CU651637.1 and OV696614.1) using "Bowtie2" (v2.4.4, [54]). Based on properly paired and mapped reads ("samtools view -f2 -q30", v1.14, [55]), the RNA-Seq fragments overlapping each gene were counted with "featureCounts" (v2.0.1, [56]). Fpkm (fragments per kilobase of transcript per million mapped fragments) values reflecting the expression level of each gene in each sample were computed with library sizes estimated using the robust method implemented in R library "DESeq2" (v1.34.0, [57]). Differential gene expression analysis between condition MB6 and BLB was also performed with "DESeq2", whose p-values were converted into q-values using R library "fdrtool" (v1.2.17, [58]). Genes associated with an estimated q-value $\leq 0.05$ and $|log2FC| \geq 1$ were called differentially expressed. Coverage by RNA-Seq fragments along the genome which served to draw transcription profiles was extracted with "bedtools genomecov" (v2.30.0, [59]) and normalized to fpkm as explained in Bidnenko *et al.* [60].

## Moron detection

The normalized gene expressions of the two replicates of *in vitro* growth in LB up to stationary phase were selected for moron detection (S1 Table, columns BLB, BR1 and BR2, log2p5fpkm). To detect morons in each prophage region, the distribution of all prophage gene normalized read counts was analyzed. It was always bimodal with a first high peak of low read counts, separated from a low peak of high read counts. We found that using a cut-off at 5-fold above the median was adapted to recover the second group of highly transcribed genes in all prophages. Genes listed as morons (i) belonged to this second group of highly transcribed genes and (ii) differed from genes needed for the phage cycle such as the master repressor.

## Homology between *E. coli* LF82 prophages and reference phages

Related phages were searched in the nr/nt nucleotide collection of the NCBI by BLASTn, within the Viruses taxid 10239 as of March 2020, using the megaBLAST parameters. For each prophage, the type phage of the viral species (as defined by the ICTV, https://ictv.global/) and

the closest phage were retained for comparison through genomic alignments. Alignments were realized using the R package GenoPlotR [61], based on tBLASTx to generate the comparison files and using a filter length of 50. Images were generated with the plot_gene_map function with the blue_red global color scheme.

### Determination of the Cyrano plasmid copy number

Relative qPCR were done on *E. coli* LF82 cultures to determine the Cyrano episome / LF82 chromosome ratio. A sample of three MAC2204 overnight cultures (LB, 37˚C, $OD_{600}$ between 3.7 and 4.0) was serially diluted in pure water 1:50, 1:100, 1:200, 1:400 and 1:800. The Luna Universal qPCR Master mix from NEB (Ref M3003E) was used with OPM 80–81 primers to quantify the LF82 chromosome copy number or JC 96–97 primers to quantify the Cyrano plasmid copy number (250 nM final each, Table 4). Nine μL of this mix were added either to 6 μL of the diluted LF82 culture samples, or 6 μL of H2O (negative control) and run in a StepOne Real-Time PCR System (ThermoFisher scientific) with the following program: 95˚C 1 min, (95˚C, 15s; 60˚C, 30s) 40 cycles, followed by melting curves. Results obtained were analyzed using the StepOne Software 2.3 and Cyrano plasmid / LF82 chromosome ratios were calculated with the ΔΔCt methodology.

### Sequencing of *E. coli* LF82 virome

One liter of LF82 culture grown under agitation at 37˚C in LB to an $OD_{600}$ ~1 was centrifuged for 7 minutes at 5,000 g at 4˚C. Supernatant was filtrated on a 0.2 μm membrane and nanoparticles were precipitated with 10% PEG 8,000 and 0.5 M NaCl during an overnight incubation at 4˚C. The preparation was then centrifuged for 30 minutes at 5,000 g and supernatant was removed. A second centrifugation for 5 minutes was added to completely eliminate the supernatant. The pellet was resuspended in 2 mL of SM Buffer (50 mM Tris-HCl pH 7.5, 100 mM NaCl, 10 mM $MgSO_4$) and treated during 30 minutes at 37˚C with 4 μg of RNAse and 2 U of Turbo DNAse (Ambion, Ref AM2239). Another incubation of 30 minutes at 37˚C with an additional quantity of Turbo DNAse (2 U) was added to maximize the removal of bacterial DNA from the sample. Then Turbo DNAse was inactivated with 10 mM of EDTA pH 8. To extract phage DNA, we performed two phenol-chloroform–isoamyl alcohol (25:24:1) extractions followed by a chloroform-isoamyl alcohol (24:1) purification step. Then DNA was precipitated with two volumes of pure ethanol at 4˚C and pelleted with a full-speed centrifugation for 5 minutes. Ethanol was eliminated by evaporation and the DNA pellet was resuspended in 40 μL of 10 mM Tris-HCl pH 8. Double-stranded DNA concentration was measured with a Qubit (dsDNA Broad range assay kit, Invitrogen, Ref Q32850) at 88 ng/μL, and 525 ng were sent to Eurofins for Illumina Hiseq paired-end sequencing (2 million read depth).

Reads obtained were filtered with TRIMMOMATIC [62] to keep only those of high quality using the command ILLUMINACLIP:TruSeq3-PE.fa:2:30:10 LEADING:3 TRAILING:3 SLIDINGWINDOW:4:20 MINLEN:125. Remaining reads were mapped with Bowtie2 (-N 0 -L 32) [54] on a sequence that concatenated the LF82 chromosome (CU65163, [22]) and the "LF82 plasmid", now Cyrano (CU638872, [22]). Finally the coverage information was extracted using Tablet [63] and represented with ggplot2 on R. Coverage corresponding to the mean genomic DNA contamination was calculated by using unmapped reads from a Bowtie2 alignment on a sequence concatenating the all 5 prophages and the bacterial region between Gally and Perceval, that is transduced by Gally (7.6 reads/bp).

We used one-sided t-tests (alternative = « greater ») to determine whether the coverage of LF82 prophages is significantly higher than the surrounding background level of contaminating bacterial DNA. For Tritos and Cartapus, prophage coverages were divided into 5 kb

portions and compared to a 100 kb portion of bacterial DNA coverage around their integration site (50 kb before *attL* and 50 kb after *attR*, also divided in 5 kb windows). For Gally and Perceval, taking into account the lateral transduction process, the prophage coverages (5 kb windows) were compared to the bacterial DNA coverage upstream of Gally (LF82: [700,000; 800,000], divided in 5 kb windows, not affected by the Gally-mediated lateral transduction).

In order to precisely delimit prophage borders, clipped reads, which mapped both on the 5' and 3' ends of the prophage and provide evidence of its recircularization, were identified thanks to Tablet. These boundaries were also verified by PCR amplification and sequencing and are reported in Table 2. The boundaries indicated contain the whole prophage, and only its *attR* site, so that it is possible to reconstitute an *attB*-like site after removing this region. An exception is made for Perceval prophage, for which we have kept its *attL* site, because its *attP* site is more similar to its *attR* than its *attL* site.

## Phage isolations

Perceval was isolated from the LF82 ΔGally strain (MAC2225), using a 5 mL exponentially growing culture treated with 2 μg/mL ciprofloxacin during 4h30. The culture was then centrifuged for 4 minutes at 11,000 g, 4˚C and filtrated on 0.2 μm membrane. As no phage plaque was observed with this supernatant on *E. coli* DH10B, an enrichment step was added: 250 μL of supernatant was adsorbed for 10 minutes, 37˚C, on 500 μL of DH10B overnight culture supplemented with 10 mM MgSO$_4$ and 1 mM CaCl$_2$. Then the mix was diluted in 50 mL of Lennox and incubated overnight at 37˚C. Small and clear phage plaques were obtained after plating 40 μL of the resulting supernatant with 100 μL of DH10B overnight culture in an agar overlay (10 g/L bactotryptone, 2.5 g/L NaCl, 4.5 g/L agar). Both plaque types were streaked for purification, and large stocks were prepared by lysis confluence on plates and recovery in SM buffer by diffusion (1 hour, 4˚C) followed by filtration (0.2 μm). PCR analysis using diagnostic primers for each predicted LF82 prophage (Table 4) gave positive results with the Perceval primers (OPM 15–16 and OFL 290–291), but not with the others, indicating that both purified phages were Perceval.

A Tritos phage plaque was isolated once, from a LF82 culture supernatant directly plated in an agar overlay with strain MG1655 *hsdR-* (MAC1403). After purification (streaking) and amplification (lysis confluence on plate), PCR analysis gave a positive result with Tritos primers (LF-ph3-port 1 and 2), but not with the others, meaning that the isolated phage was Tritos.

Two Gally-Perceval hybrid phages (named Galper1 and 2) were also isolated from the supernatant of LF82 cultures, after plating in an agar overlay with either LF82 ΔGally (MAC2225) or MG1655 *hsdr-* (MAC1403) strains. In each case, a single phage plaque was obtained (small and clear), streaked, and amplified in liquid cultures with their strain of isolation. For Galper1, in order to remove the PCR signal from LF82 DNA contamination, the crude lysate was treated with DNAse: 10 μL of the phage stock was diluted 1:100 in H$_2$O and treated with 1 U Turbo DNAse (Ambion, Ref AM2239) for 1h30 at 37˚C. DNAse was then inactivated with a 30 minutes incubation at 95˚C. For each phage isolation, PCR with the diagnostic primers (Table 4) were positive for both OPM7-OPM8 (targeting Gally) and OFL290-OFL291 (targeting Perceval), indicating that these phages were composed of parts of Gally and Perceval genomes.

All conditions tested to isolate Gally plaques were unsuccessful. These included: (i) plating LF82 supernatants on various strains (LF82 ΔGally (MAC2225), C+ (MAC2266), TD2158 (MAC2294) and MG1655 *wbbl+* (MAC2310)), (ii) plate incubation at different temperatures (25, 30, 37 and 42˚C) and (iii) Gally enrichment on TD2158 or MG1655 *wbbl+*, as follows: 250 μL of LF82 supernatant was adsorbed for 10 minutes at 37˚C on 500 μL of overnight

MAC2294 or MAC2310 cultures supplemented with 10 mM $MgSO_4$, 1 mM $CaCl_2$, then diluted in 50 mL of Lennox supplemented with 10 mM $MgSO_4$, 1 mM $CaCl_2$ and incubated overnight at 37˚C. However, Gally was able to lysogenize a LF82 ΔGally strain (MAC2225). To test this, a phage stock produced from a Gally KanR derivative (Gally Δ*recT-gam-abc1*::KanR, strain MAC2222) was incubated 30 minutes at 37˚C with LF82 ΔGally (MAC2225) at $OD_{600}$ ~1, using various MOI (0.1 and 1, phage genome quantities estimated by qPCR, final volume 1 mL). Bacteria were then centrifuged to remove unabsorbed virions (7 minutes, 5,000 g, room temperature), and bacterial pellets were resuspended in 1 mL of Lennox at 37˚C, serially diluted, plated in 5 mL agar overlay on 25 mL of Lennox agar plates, and incubated at 37˚C during 1h30, in order to allow expression of the kanamycin resistance gene. Next, to select for KanR lysogens, a second agar overlay supplemented with kanamycin (100 μg/mL final concentration for the entire volume of the plates) was added. After a 24h incubation at 37˚C, KanR and viable counts were estimated and lysogenization frequencies calculated by the ratio of KanR over total bacteria. Background frequency of KanR mutants was below $7x10^{-9}$.

## Observation of virions by electron microscopy

1 mL of purified stocks of Tritos ($1.4x10^7$ PFU/mL), Perceval ($6.2x10^{10}$ PFU/mL) and Galper1 ($10^{11}$ PFU/mL) phages were concentrated for TEM observation, by successive washes in ammonium acetate following the protocol from Nicolas Ginet (CNRS, France, personal communication). After a 1-hour centrifugation (20,000 g, 4˚C), pellets were resuspended in 1 mL of 0.1 M ammonium acetate pH 7 (previously filtrated on 0.2 μm membrane). Tubes were centrifuged for 1 further hour and pellets resuspended in 50 μL of 0.1 M ammonium acetate pH 7.

For Gally imaging, 200 mL of LF82 overnight culture were centrifuged for 7 minutes at 5,200 g. Supernatant was filtrated on 0.2 μm membrane, and centrifuged for 3h at 143,000 g, 4˚C to concentrate the virions. Resulting pellet was resuspended in 12 mL of 0.1 M ammonium acetate pH 7, before being centrifuged once again for 2 hours at 154,000 g, 4˚C. The final pellet was resuspended in 30 μL of 0.1 M ammonium acetate pH 7.

Cyrano was visualized by TEM using the same protocol as that used for the observation of Gally, except that the LF82 culture ($OD_{600}$ ~0.3) was treated for 2 hours with 0.09 μg/mL ciprofloxacin.

Ten μL of each virion preparations were absorbed onto a carbon film membrane placed on a 300-mesh copper grid and stained with 1% uranyl acetate dissolved in distilled water. After drying at room temperature, grids were observed with Hitachi HT 7700 electron microscope at 80 kV (Elexience–France) and images were acquired with a charge coupled device camera (AMT). Finally, tails and capsids were measured using ImageJ software [64].

## Gally-Perceval hybrid genome assembly

Galper1 was entirely sequenced following the same first steps described above for the virome sequencing. After read cleaning, a dereplication step was computed, using the USEARCH9 command line -fastx_uniques [65], pairs were reconstituted using FASTQ_PAIR and reads were assembled with SPADES (—careful -k 21,33,55,77,99,127 option, [66]). A single contig of 44,690 bp, corresponding to the complete genome of Galper1, was obtained (S2 Fig).

To test whether the recombination junctions were placed similarly in Galper2, these two regions were PCR amplified with OPM 50–51 and OPM52-53, and sequenced (S2 Fig).

## Determination of minimal inhibitory concentrations of antibiotics for LF82

We first determined the ratio between $OD_{600}$ and CFU/mL for LF82 (MAC2218). $OD_{600}$ from three independent 18 hours cultures of LF82 were measured, and samples were plated and

incubated overnight at 37˚C. Colony counts indicated that a saturated LF82 culture contains about $9.6 \times 10^8$ bacteria/mL per OD unit.

Taking this ratio into account, we then determined the minimal inhibitory concentrations (MIC) of LF82 for gentamicin (Sigma, ref G1264-1G, resuspended in $H_2O$), cefotaxime (Sigma, ref 219380, resuspended in $H_2O$), trimethoprim (Sigma, ref T7883-5G, resuspended in 100% DMSO) and ciprofloxacin (Sigma, ref 17850-5G-F, resuspended in 100 mM HCl) in Lennox medium by following the protocol from Wiegand *et al.* [67]. Three independent 18 hours cultures of LF82 were diluted to $10^6$ CFU/mL and 1 mL of each was added to 1 mL of Lennox containing increasing concentrations of antibiotics (two-fold steps): $6.3 \times 10^{-2}$ to 125 µg/mL for gentamicin, $9.4 \times 10^{-4}$ to $9.6 \times 10^{-1}$ µg/mL for cefotaxime, $2.3 \times 10^{-3}$ to $2.4 \times 10^{-1}$ µg/mL for ciprofloxacin and $7.8 \times 10^{-3}$ to 8 µg/mL for trimethoprim. To verify the input bacterial titer, cultures containing no antibiotics were numerated on Lennox agar and incubated overnight at 37˚C. Cultures with antibiotics were incubated at 37˚C for 20 hours under agitation. MICs corresponded to the minimal concentrations of antibiotics that completely inhibits growth of LF82 ($OD_{600}$ below 0.05): 15.63 µg/mL for gentamicin, 0.72 µg/mL for cefotaxime, 0.09 µg/mL for ciprofloxacin and 0.33 µg/mL for trimethoprim.

To confirm these results for microplate cultures, 50 µL of antibiotics dilutions were added to 50 µL of diluted LF82 (MAC2218) overnight culture ($10^6$ CFU/mL) and plated in a 96-wells plate which then was closed with a semi-permeable filter (Gas permeable film, 4titude, Ref 4ti-0516/96) to prevent evaporation. The plate was incubated for 20 hours at 37˚C in a Tecan fluorimeter. Using the TECAN I-CONTROL software, $OD_{610}$ of each well was measured every 3 minutes after 15 seconds of orbital shaking of the plate at 158.9 rpm and a wait time of 5 seconds. As previously done, 10 µL from cultures without any antibiotics were collected in order to verify the input bacterial titer. MIC values obtained from microplate cultures were similar to those obtained in tubes: 15.63 µg/mL for gentamicin, 0.24 µg/mL for cefotaxime, 0.06 µg/mL for ciprofloxacin and 0.125 µg/mL for trimethoprim.

## Quantitative PCR of LF82 phage concentration in uninduced and induced *in vitro* conditions

Overnight culture of LF82 (MAC2218) was diluted 1:500 and grown at 37˚C to an $OD_{600}$ between 0.2 and 0.3. Cultures were then diluted 1:2 in Lennox, with or without antibiotics at the MIC, and 200 µL of each dilution was placed in a 96-wells plate. The plate was closed with a semi-permeable membrane and incubated for 2 hours at 37˚C in a Tecan fluorimeter, until the cultures without antibiotics reached a plate-reader $OD_{610}$ between 0.25 and 0.3. To recover phage supernatants, the plate was centrifuged for 7 minutes at 5,200 g at 4˚C and supernatants were filtrated on 0.2 µm membrane. 107 µL of samples with similar growth profiles and final ODs were treated during 1 hour at 37˚C with 2 U Turbo DNAse to remove the bacterial DNA. This was followed by a 30-minutes incubation at 95˚C to inactivate the DNAse and release phage DNA from virion capsids. Final samples were diluted 1:50 and 1:100 in pure water, and 6 µL were used for the PCR quantification.

The bacterial DNA of LF82 was used as a reference point for qPCR measurements of phage copy number (1 prophage copy per genome). Genomic DNA was extracted from an overnight culture lysate treated twice with phenol-chloroform–isoamyl alcohol (25:24:1), followed by four chloroform-isoamyl alcohol (24:1) extractions. DNA was ethanol precipitated and resuspended in 10 mM Tris-HCl pH 8. DNA was quantified with Qubit (dsDNA Broad range assay kit, Invitrogen, Ref Q32850) and serially diluted in 10 mM Tris-HCl pH 8 to obtain a range from about 50 to $5.10^5$ copies of *E. coli* LF82 genome per 6 µL.

To quantify the packaged phage DNA, we used the Luna Universal qPCR Master mix from NEB (Ref M3003E) with primers described in Table 4 at 250 nM each, to target specifically

phage genomes or to target the yersiniabactin biosynthesis salycil-AMP ligase protein encoding gene (*ybtE*) from LF82 in order to evaluate the bacterial DNA contamination of our samples. Nine μL of this mix was added either to 6 μL of diluted viral samples, 6 μL of LF82 genome for the qPCR reference, or 6 μL of $H_2O$ (negative control) and run in a StepOne Real-Time PCR System (ThermoFisher scientific) with the following program: 95˚C 1 min, (95˚C, 15s; 60˚C, 30s) 40 cycles, followed by melting curves. Results obtained were analyzed using the StepOne Software 2.3.

## Survival of *E. coli* LF82 in the presence of ciprofloxacin

Overnight cultures of LF82 (MAC2204) and LF82 ΔGally (MAC2225) were diluted 1:500 in Lennox and incubated at 37˚C to an $OD_{600}$ between 0.2 and 0.3. 40 μL of Lennox broth supplemented or not with ciprofloxacin (0.09 μg. $mL^{-1}$ final concentration) was added to 160 μL of each culture. The $OD_{600}$ of the cultures was monitored in a 96-well plate closed with a semi-permeable membrane for 5 hours at 37˚C in a Tecan fluorometer.

## *E. coli* LF82 survival in macrophage

THP1 (ATCC TIB-202) monocytes ($4.75x10^5$ cells/mL) were differentiated into macrophages in phorbol 12-myristate 13-acetate (PMA, 20 ng/mL). *E. coli* LF82 (MAC2204), *E. coli* ΔGally (MAC2225) or *E. coli* Δ*mnt* (MAC2774) were used to infect $4.75x10^5$ THP1 macrophages. After 1, 6 or 24 hours Post-Infection (P.I.), THP1 macrophages were lysed with 500 μL of 1% Triton-PBS. Lysate was plated on Lennox agar medium and incubated overnight at 37˚C. Colonies were counted to determine the CFU/mL after macrophage infection of each LF82 strain at each time point.

## Analyses of the Gally phage induction *in vitro* by epifluorescence microscopy

For tracking MCP-GFP expression on a pad of minimal agar medium, an overnight culture in Lennox at 37˚C of the MAC2606 strain was diluted 100-fold in fresh medium. At $OD_{600}$ ~0.3, 0.09 μg/mL of ciprofloxacin was added and cells were deposited on a slide covered with 1.5% agarose in M9 minimal medium supplemented with 0.2% glucose and 0.09 μg/mL of ciprofloxacin. Cover slips were positioned and slides were examined at different time points until 160 min at 37˚C with the Carl Zeiss AxioObserver Z1 fluorescent microscope. Images were acquired with a 100x oil immersion objective and the Zen software (Carl Zeiss). Time-Lapse Image analysis was performed as follows: bacteria were segmented first with Omnipose [68] and tracked with Trackmate [69]. Finally, time-point measurements were performed with FIJI/ImageJ [70].

MCP-GFP expression in Lennox medium was analyzed on exponential growth cells (MAC 2606 strain) 60 min after addition or not of 0.09 μg/mL of ciprofloxacin. Cells were examined on a slide covered with 1.5% agarose in M9 minimal medium as described above and images were analyzed by counting dotted fluorescent cells (almost all fluorescent cells grown in Lennox medium supplemented with ciprofloxacin contained fluorescent foci) and non-fluorescent cells using the Image J software.

## Gally prophage induction in macrophage, followed by epifluorescence microscopy and qPCR

Strains OEC2481 and OEC2425 inside macrophages were observed as follows: OEC2481 and OEC2425 strains were inoculated in Lennox medium and incubated at 37˚C at 180 rpm. The overnight bacterial culture was diluted 100-fold in fresh medium. Once $OD_{600}$~0.5 was

obtained, macrophages THP1 (see above the monocytes differentiation protocol and infection) were infected with 119 μL of the bacterial culture, and incubated at 37˚C, 5% $CO_2$ as described [71]. After 40 min, 6 hours and 24 hours P.I., macrophages were fixed with formaldehyde 3.7% (Ref: F8775 Sigma-Aldrich) for 30 minutes at room temperature, and washed twice with PBS. Then the lamella was mounted with Dako. Imaging was performed on an inverted Zeiss Axio Imager with a spinning disk CSU W1 (Yokogawa) at 63X magnification. Metamorph Software (Universal Imaging) was used to collect the data.

The production of Gally and Cyrano phages in macrophage was quantified by qPCR as follows: 1.2 to $1.7x10^7$ macrophages THP1 were infected with 252 μL of the bacterial culture of the MAC2204 strain as described above. We verified that LF82 virions produced *in vitro* did not contaminate our assay by washing the bacteria before the macrophage infection: no significant difference in the amount of Gally phages was detected in macrophages after infection with bacteria previously washed in Lennox or not. Six hours P.I., macrophages were lysed with Triton 0.075% for 10 min at room temperature. Lysed macrophages were then scraped from the culture well, filtered on a PES-membrane of 0.2 μm and stored at 4˚C until the next step, 12 hours later. Viral particles were then concentrated 10-fold with 10% PEG 8,000 and 0.5 M NaCl (see virome sequencing section), treated with 2 U of Turbo DNAse, diluted in pure water and quantified with the Luna Universal qPCR Master mix from NEB as previously described.

We tested whether the virions produced in macrophages were quantitatively recovered after cell lysis, and remained in the macrophage lysate until precipitation 12 hours later. For this, a spike-in of ~$10^8$ Gally virions (300 μL of 10-fold concentrated LF82 supernatant) was added in the macrophage cultures (6h P.I.) before their lysis, or after the lysate filtration. We observed a 18 to 20-fold decrease in the expected Gally virions concentration when the spike-in was added before macrophage lysis and until its analysis by qPCR. Even when added after lysate filtration, Gally virions decreased by a 8 to 13-fold upon the 12 hours storage at 4˚C. Therefore, most of Gally virions instability occurs during macrophage lysate storage. To take in account this instability, a 20-fold multiplication correction was applied to all Gally virion quantifications in macrophages. Following the same protocol, we determined a loss factor of only 3 for Cyrano, after filtration of the lysate and until its analysis by qPCR, which means that Cyrano is more stable than Gally in this medium.

## Semi-quantitative PCR on the *ybtE* gene and *attL*, *attB* and *attP* sites

LF82-infected macrophages were lysed 6 hours P.I. as described above, centrifuged at 5,200 g for 7 min, then washed in PBS 1X. Bacterial and infected macrophage DNA was purified using the PureLink Genomic DNA Mini Kit (Invitrogen K182001). As controls, DNA from LF82 bacteria grown in Lennox to $OD_{600}$ = 0.3 and then treated or not for 1 hour with ciprofloxacin (0.09 μg/mL) was purified using the Wizard Genomic DNA Purification Kit (Promega). Amplification of the *ybtE* gene and *attL*, *attB*, and *attP* sites was performed by PCR with the following program: 94˚C 30s, (94˚C, 30s; 50˚C, 30s, 68˚C, 45s) 30 cycles using OneTaq polymerase (NEB) and the oligonucleotide pairs OPM80/OPM82, OPM75/Maj281, JC206/JC207, and Maj280/Maj281, respectively. PCR products were analyzed by 1X TBE 1.5% agarose gel electrophoresis in presence of ethidium bromide. After migration, bands were revealed by the BioRad ChemiDoc MP imaging system and quantified with the Image Lab software (BioRad) using known DNA quantity of bands from the DNA ladder (SmartLadder, Eurogentech).

## Genome and reads submissions

The re-annotated genomes of the phages are available from the European Nucleotide Archive browser (http://www.ebi.ac.uk/ena/browser/view) with the following accession numbers:

OV696608 for Gally, OV696612 for Perceval, OV696610 for Tritos, OV696611 for Cartapus and OV696614 for Cyrano. Raw data obtained from the virome sequencing have been deposited (accession number: ERR8973296).

## Supporting information

**S1 Fig. Cyrano, the phage-plasmid of *E. coli* LF82.** A. Whole genome comparison of the LF82 phage-plasmid Cyrano and SSU5. A tBLASTx comparison was performed and visualized with the R package Genoplot. The heat map and gene color indications used here are the same as those used for Fig 1. B. Determination by qPCR of the Cyrano copy number per *E. coli* LF82 bacteria. Each dot corresponds to one biological replicate. The mean of these values (5.5) is represented by a vertical line.
(TIF)

**S2 Fig. Genetic analysis of two Gally-Perceval hybrids, Galper1 and Galper2.** A. Genetic map of the Galper hybrids. Grey triangles indicate the two recombination endpoints between Gally and Perceval. B. Transmission electron microscopy photograph of the purified Galper1. Scale bar is 50 μm long. C and D. Sequence analysis of the first (C) and the second (D) recombination endpoints in Galper1 (upper panels, red) and Galper2 (bottom panels, blue), which occur respectively in a 256 and a 179 bp region of partial homology between Perceval and Gally ((C) 84% identity, (D) 72% identity).
(TIF)

**S3 Fig. Impact of the Gally Mnt repressor on LF82 survival in macrophage at 1, 6 and 24 hours P.I.** Each dot corresponds to a biological replicate, from an independent macrophage infection. Horizontal black lines represent mean values.
(TIF)

**S4 Fig. Monitoring of MCP-GFP expression in exponentially growing LF82 cells (MAC2606) plated on minimal medium supplemented with ciprofloxacin at the MIC.** A. Fluorescence (GFP) and phase channel images obtained at different time points after deposition are shown, along with an overlay of these images. Red arrows: cells becoming fluorescent and lysing during incubation. White arrow: lysed cell without MCP-GFP induction. B. Quantification of the four categories of cells monitored: intact or lysed non-fluorescent cells (solid or broken gray lines, respectively), and intact or lysed fluorescent cells (solid or broken black lines, respectively). C. Cell lysis over time as a function of the prior induction (black dashed line) or not (gray dashed line) of MCP-GFP fusion protein expressed from Gally phage.
(TIF)

**S5 Fig. Confocal imaging of THP-1 macrophages at 40 minutes P.I. with LF82-pP*rpsm*-*mCherry* (OEC2425) (left panel) and LF82-pP*rpsm*-*mCherry* Gally *mcp-GFP* (OEC2481) (right panel).**
(TIF)

**S6 Fig. Confocal imaging of THP-1 macrophages at 24 hours P.I. with LF82-pP*rpsm*-*mCherry* (OEC2425) (left panel) and LF82-pP*rpsm*-*mCherry* Gally *mcp-GFP* (OEC2481) (right panel).**
(TIF)

**S7 Fig. Estimation of Gally replication, excision and recircularization in different growth conditions.** A. PCR amplification product analyzed by gel electrophoresis in the presence of ethidium bromide, obtained with the following oligonucleotide pairs: OPM80/OPM82 (*ybtE*),

OPM75/Maj281 (*attL*), JC206/JC207 (*attB*) and Maj280/Maj281 (*attP*). The DNA templates used (quantities indicated below the gels) were purified from LF82 bacteria grown either in Lennox medium, Lennox with ciprofloxacin (at the MIC) for ~1 hour, or within macrophages for 6 hours, as indicated. Two replicates were analyzed for each condition. M: molecular weight marker. The asterisk denotes a contaminant amplified product obtained with the DNA template from LF82 bacteria grown in macrophages. B. Evaluation of the impact of contaminant amplification (*) on the amplification of the Gally *attB* site. The excision site was amplified from mixes of the indicated amounts of purified LF82 DNA template extracted from bacteria grown in macrophages (DNA mac.) or Lennox (DNA Lennox) and analyzed by agarose gel electrophoresis. Amplification of the contaminant product does not repress amplification of the *attB* site. C. Ratio of the different *att* sites over the bacterial *ybtE* gene, used as a reference, in unstressed (white bars) or stressed (ciprofloxacin, black bars) *in vitro* growth conditions and in macrophages (6 hours P.I., grey bars). Bands on the gel were quantified using Image Lab software for both sets of replicates. Except for some *attB* ratios, ratios were computed by dividing *att* site amounts generated from a defined input of template DNA, by *ybtE* amounts generated from the same DNA input. For *attB* under Lennox (+/- cip) growth conditions, the *attB*/*ybtE* ratio was calculated by dividing the amount of *attB* PCR products obtained from 5, 1, and 0.2 ng of template DNA by the amount of *ybtE* amplified product from 0.62, 0.16, and 0.04 ng, respectively. The ratios obtained were then multiplied by the difference in the amount of template DNA used for PCR for *attB* and *ybtE*.
(TIF)

**S1 Table. Raw data extracted from S4 Table concerning only the five prophage regions (one page per prophage), after growth of LF82 in LB medium, DMEM and macrophage conditions (from [27]).** For LB data, the analysis leading to moron identification is shown.
(XLSX)

**S2 Table. qPCR data obtained with *in vitro* samples and after macrophage infections, and phages/bacteria ratios calculated from *in vitro* cultures (grown in Lennox or Lennox+-ciprofloxacin at the MIC) or after macrophage infection (6h P.I.).**
(XLSX)

**S3 Table. Summary of the results obtained from the characterization of *E. coli* LF82 phages in this study.** (+)*: virions quantification contaminated by Gally-mediated lateral transduction. ND: not determined.
(PPTX)

**S4 Table. Analysis of the RNA-Seq raw data from GEO accession GSE154648 (from [27]).** Legend is included in the table.
(XLSX)

## Acknowledgments

We are grateful to Christine Longin (MIMA2 platform) for her help with the TEM observations, to Alice Eon-Bertho for technical help, to Julien Lossouarn for the phage genomes and reads submissions and to the Migale platform (INRAE) for the bio-informatics environment.

## Author Contributions

**Conceptualization:** Marianne De Paepe, Marie-Agnès Petit, Olivier Espeli, François Lecointe.

**Data curation:** Pierre Nicolas, Goran Lakisic, François Lecointe.

**Formal analysis:** Pauline Misson, Emma Bruder, Jeffrey K. Cornuault, Marianne De Paepe, Marie-Agnès Petit, Olivier Espeli, François Lecointe.

**Funding acquisition:** Marie-Agnès Petit.

**Investigation:** Pauline Misson, Emma Bruder, Jeffrey K. Cornuault, Marianne De Paepe, Gaëlle Demarre, Marie-Agnès Petit, Olivier Espeli, François Lecointe.

**Methodology:** Pauline Misson, Emma Bruder, Jeffrey K. Cornuault, Marianne De Paepe, Pierre Nicolas, Gaëlle Demarre, Goran Lakisic, Marie-Agnès Petit, Olivier Espeli, François Lecointe.

**Project administration:** François Lecointe.

**Resources:** Marie-Agnès Petit, François Lecointe.

**Software:** Pauline Misson, Emma Bruder, Marianne De Paepe, Pierre Nicolas, Goran Lakisic, Marie-Agnès Petit, Olivier Espeli, François Lecointe.

**Supervision:** François Lecointe.

**Validation:** Pauline Misson, Emma Bruder, Jeffrey K. Cornuault, Marianne De Paepe, Pierre Nicolas, Goran Lakisic, Marie-Agnès Petit, Olivier Espeli, François Lecointe.

**Visualization:** Pauline Misson, Emma Bruder, Jeffrey K. Cornuault, Marianne De Paepe, Pierre Nicolas, Goran Lakisic, Marie-Agnès Petit, Olivier Espeli, François Lecointe.

**Writing – original draft:** Marie-Agnès Petit, François Lecointe.

**Writing – review & editing:** Pauline Misson, Emma Bruder, Jeffrey K. Cornuault, Marianne De Paepe, Marie-Agnès Petit, Olivier Espeli, François Lecointe.

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
