## [Decision Letter · Decision Letter 0]

23 Jun 2022

Dear Dr Lecointe,

Thank you very much for submitting your manuscript "Prophage taming by the adherent-invasive Escherichia coli LF82 upon macrophage infection" for consideration at PLOS Pathogens. As with all papers reviewed by the journal, your manuscript was reviewed by members of the editorial board and by several independent reviewers. In light of the reviews (below this email), we would like to invite the resubmission of a significantly-revised version that takes into account the reviewers' comments.

We apologize for the very long time it took to secure reviews for this work. Unfortunately three independent scientists have reviewed the work and all have significant concerns. The associate editor has also reviewed the manuscript and concurs with some but not all of the reviewer's criticisms. We cannot accept the manuscript as is but would be willing to consider a substantially revised work that addresses the reviewers primary concerns.

The associate editor does not feel that one comment from reviewer 1 needs to be addressed directly and that there may be some confusion in that reviewer's logic. The fact that a Gally knockout in bacteria is equivalent to a repressed prophage makes sense and is not contradictory. The associate editor does agree that a shortcoming is a lack of a clearly proposed mechanism for how prophages are repressed in the macrophage. Reviewer 3 has several concerns about methodologies and controls that need to be addressed.

We cannot make any decision about publication until we have seen the revised manuscript and your response to the reviewers' comments. Your revised manuscript is also likely to be sent to reviewers for further evaluation.

Sincerely,

William Navarre

Associate Editor

PLOS Pathogens

Raphael Valdivia

Section Editor

PLOS Pathogens

Kasturi Haldar

Editor-in-Chief

PLOS Pathogens

orcid.org/0000-0001-5065-158X

Michael Malim

Editor-in-Chief

PLOS Pathogens

orcid.org/0000-0002-7699-2064

We apologize for the very long time it took to secure reviews for this work. Unfortunately three independent scientists have reviewed the work and all have significant concerns. The associate editor has also reviewed the manuscript and concurs with some but not all of the reviewer's criticisms. We cannot accept the manuscript as is but would be willing to consider a substantially revised work that addresses the reviewers primary concerns.

The associate editor does not feel that one comment from reviewer 1 needs to be addressed directly and that there may be some confusion in that reviewer's logic. The fact that a Gally knockout in bacteria is equivalent to a repressed prophage makes sense and is not contradictory. The associate editor does agree that a shortcoming is a lack of a clearly proposed mechanism for how prophages are repressed in the macrophage. Reviewer 3 has several concerns about methodologies and controls that need to be addressed.

Reviewer's Responses to Questions

**Part I - Summary**

Reviewer #1: In this manuscript, the authors thoroughly characterized 5 prophages in the adherent-invasive E. coli LF82. The numbers of viral genomes in the culture supernatant were determined, and the morphology was also determined for four phages (except for Cartapus). For Gally, the most highly induced prophage, evidence for lateral transduction was observed. The deletion of Gally did not affect bacterial survival during the infection of macrophages. Contrary to the expectation, the induction of prophages, including Gally, was reduced in the macrophages. In the case of Gally, some parts of the prophage genome appeared repressed during the infection of macrophages. Based on these results, the authors concluded that the E. coli strain actively controls the induction of the most active prophage Gally, which might enhance the bacterial survival in macrophages.

This manuscript is well-written with high clarity. In general, science is very solid, and I do not have any serious concerns about the methodology. However, it is difficult to accept the authors’ conclusion. The authors demonstrated that Gally plays no significant role in the bacterial survival in macrophages. Then, how could reducing induction of the prophage help bacteria survive better in macrophages? Since the induction of other prophages is also reduced, it is possible that the intracellular environment of macrophages has a low inducting activity for the prophages, rather than the LF82 strain actively reducing the prophage induction? Finally, no reasonable mechanistic model was provided for the selective repression of Gally genes.

Reviewer #2: In this study, the authors performed detailed analyses of 5 active prophages of E. coli strain LF82, the prototype AIEC strain. They isolated two phages, visualized the 4 phages, and identified moron genes. As for the most active one (named Gally), they provide evidence of lateral transduction and formation of hybrid phages and show that Gally (and other phages) do not produce much lower numbers of virion in macrophages. I appreciate the authors' efforts, especially those to isolate and visualize the five phages and to provide clear evidence of lateral transduction of Gally. But, a weak point of this study is that, regarding the suppression of prophage in macrophages, it is not clear what happened in macrophages, although I understand this is a difficult challenge because the prophage induction that the authors tried to analyze occurred only in a very small portion of bacterial cells.

Reviewer #3: The authors report on experiments aimed at characterizing 5 prophages in the genome of an AIEC strain. They use HT sequencing, transcriptomics, microscopy and a fluorescent reporter of lytic gene expression to examine liberation, phage lifestyle and morphology in response to antibiotics and macrophage infection. Unfortunately, the narrow range of methodologies applied, the lack of controls and experimental description make it impossible to validate the very compelling proposal that titles the paper, namely that prophage liberation is tamed upon macrophage infection.

**Part II – Major Issues: Key Experiments Required for Acceptance**

Reviewer #1: 1. To validate the conclusion, the authors need to show the relevance of Gally induction to the survival of the bacterium.

2. Also, the authors need to provide experimental evidence that bacteria, not the intracellular environment of macrophage, control the transcription of Gally.

Reviewer #2: The authors concluded that prophage induction is suppressed in macrophages. This is the major and most important finding of this study. But, as mentioned above, it is not clear what happened in phagosomes of macrophages. I have two major concerns related to it. Both are particularly important if the authors consider a mechanism/phenomenon similar to the active lysogeny observed for a Listeria prophage, phi10403S (P18, the last line~P19, L1-L4, L7-L11).

1. Virion production was investigated, but the authors did not examine the excision and replication of Gally genome, which are important hallmarks of prophage induction.

2. Regarding the interpretation of RNA-seq data interpretation, it is difficult for me understand how the up or down-regulated gene clusters are related to the transcription units of Gally. I think the authors can show how the transcription units are organized in Gally in its lytic cycle.

Ideally, Cyrano should be analyzed in parallel as this phenomenon does not seem to be Gally-specific, although analyses of Cyrano is not so easy due to its low induction and low gene expression.

Reviewer #3: Additional comments:

1. What is the rationale for the use of a 5-fold increased expression above local genes as an identifier for morons? It seems that the authors should also determine whether these genes are part of other lysogenic gene clusters/operons and whether their expression is truly independent of other phage genes to identify them as morons.

2. In the first results section the authors state “Inspection of attachment sites of the four integrated prophages did not reveal any particular gene inactivation pattern”. Can they be more specific here? What were they looking for? What are the attachment sites?

3. In Figure 2, in order to characterize phage release, the authors sequence DNase treated supernatants where they have destroyed capsids. The results for Gally are very convincing, however the results arguing that the other phages are spontaneously released are less so. A much narrower range of chromosomal background reads is shown for the other phages, and the phage signals appears to be much weaker. These things combined make it difficult to Perceval, Cartapus and Cyrano phages are present in the supernatant from cultures of E. coli LF82.

4. The authors use a previous Tn-seq analysis to estimate the copy number of Cyrano. Some empirical work must be done to validate these conclusions, as other explanations besides copy number could impact the frequency of Tn insertions in this DNA.

5. Figure 3: The authors report a relatively high amount of contaminating chromosomal locus DNA from the ybtE gene on p. 9 of the manuscript, however this control is not included in Figure 3, where the authors investigate the impact of antibiotics on phage DNA release into the supernatant. This control needs to be included to validate the conclusions/results.

6. How was the indicator strain MAC1403 identified? No mention of this is made in the results section but this is important for evaluating the veracity of the phage isolated from this strain and visualized by TEM. How do the authors know which phages these are? Could they be phages liberated from the indicator strain?

7. The description of how the Galper 1 and 2 phages should be expanded in the results section. How were these hybrid phages isolated? Could they be plaque purified and propagated? How often does hydridization between Gally and Percival happen?

8. It seems from the way some of the results are described that the authors presume the identify of the phages seen in the supernatants based on predictions related to their genomes, and not actually on any empirical identification criteria. A more detailed description of the work flow used to validate the identity of the phages, which underlies the conclusion that all of the phages are liberated under unstressed conditions and is a main point of the paper, must be included when the authors are describing their results and making conclusions.

9. Figure 5. The authors present hypotheses about the mechanisms of Gally phage replication and propagation and how it may impact the Perceval phage based on sequencing read coverages and changes in the detection of Perceval phage in a strain deleted for Gally phage. The data supports their hypothesis/model, but more experimental work is required to definitively prove the mechanism of replication proposed. The impact of deleting Gally phage on Perceval yield should be compared side-by-side between the wild-type and Gally deletion mutant in the same experiment.

10. The speculation about the mechanism of induction of the phages by antibiotics requires more experimental validation. Why do the authors not investigate the classic induction drug mitomycin C in recA positive and negative backgrounds to confirm their ideas?

11. Bottom p 15, the authors state that their conclusions about Gally prophage liberation in vitro vs. in macrophages rely on the assumption that the macrophage condition does not affect phage recovery. This should be empirically tested.

12. Fig. 7. The authors employ transcriptome analysis across the Gally genome in vitro vs. in macrophages to make conclusions about the viral lifestyle. What is the error in these experiments? Which changes in this figure are significant? Statistical significance is important given there are really no other empirical tests to validate this data.

13. The authors use a fusion between a viral capsid protein and GFP to visualize induction of lytic growth, however the results are not convincing. The GFP signal is quite diffuse. Further, no work is presented to validate that GFP levels/expressing cells correlate with viral production.

**Part III – Minor Issues: Editorial and Data Presentation Modifications**

Reviewer #1: 1. It would be beneficial to show whether the Gally deletion enhances the bacterial growth rate in vitro. If it does not significantly affect the overall growth rate of the bacteria, it probably will not affect the survival of E. coli in macrophages regardless of phage induction rate.

2. Pages 2, 5, and 11: “.. show that all of them form virions”, “We show that the five prophages form virions in vitro.”

: The authors failed to detect Cartapus virions. So then I wonder how they concluded that “the five LF82 prophages are induced and form virions.”

3. Page 11: “Gally could not be propagated under all conditions tested.”

: Is it possible that Gally does not lyse cells upon induction?

4. Page 5: explain what PhoP is.

5. Pages 7,9,25: NMDM -> DMEM?

6. Table 1 and page 9: Bor transcription was increased 50-fold above background in LB but not listed in Table 1. However, the table title says it lists prophages genes showing 5-fold or higher expression in LB OR DMEM, not AND.

7. Fig. 2 and 5: the light gray parts are hard to see.

Reviewer #2: P5, L13-L21(related to ref. 15): In this case, lambda is induced in macrophages in a RecA-independent / phoP-dependent mechanism (by membrane stress induced by the mCramp1 anti-microbial peptide, ). It is not clear for me how the authors consider such kinds of stresses as potential factors for prophage induction.

P7, L3 and P24 (prophage content of LF82): Does this strain contain degraded prophages/prophage remnants? How many prophage-like regions were predicted by PHASTER? In the encapsidated DNA samples, did the authors detect significant numbers of reads from some of these regions?

P7, L15-17: As the authors reannotated the 5 prophages, please provide a supplementary table to show the RNA seq data for the 5 prophage regions, including the "local background" of each prophage region and the data in macrophages. I looked at the data presented in Supplementary Data 2 in Ref. 26, but it is not easy to understand how the current results are related to the data presented in the Supplementary Data 2.

P10, L14-L15 (concentration of Gally genome): Considering the chromosome DNA contamination level, the actual encasidated Gally genome seems to be much lower.

P14, L2-3 (antibiotics treatment, 2 hr induction): Why did the authors not use MMC? Was MMC treatment less effective in the induction of LF82 prophages? Why and how did the authors decide to employ this condition (2 hr treatment)? Clear cell lysis or some other signs of phage induction were observed at this time point? I feel that 2 hr is too short to achieve the maximum induction level.

P15, 10 (leftward region) and Figure 7: The region from c2 to the last gene before xis is not a replication module. Related to this, although the gene map of Gally is presented in Figure 1, please show it (the gene organization as "prophage") in this figure to make it easier for readers to understand the data in Figure 7 in the context of gene organization.

P15, L13-L18 (mRNA level of regulators): In both samples, phage induction occurred only in a very small fraction of bacterial cells. It seems that the authors do not consider this point, at least here.

P15, the second last line (2 to 5x106): I can not understand how this value was obtained.

P25, L19: Does the SOS response in macrophages peaks at 6-hr P.I.? In my understanding (according to Ref. 24), SOS response starts at around 6-hr P.I.

Additional more minor comments:

P4, L5 & L10: Refs 2-5 are adequate and required for "up to 70% of strains in species of the human microbiota"? Ref. 10 is related to "metabolic genes"? Refs 11 & 12 (identification of lom and bor) are adequate for "adaptation to a given environment"?

P7, L5 (prophage names): Why and how did the authors give specific names to the five prophages? This is rather a personal question, but interesting for me.

P8, Table 1 title (or): Should be "and".

P9, the last line (180kb): Please explain what this is.

P7, L10-11,14 (HK620, P22, Fels2): Please cite appropriate references for the readers who are not so familiar with these phages.

P11, L2 & P17, L18-L20 (failure in the isolation of Gally and infectivity of Gally): Can Gally adsorb to LF82 cells? As the Gally-deleted LF82 is not sensitive to Gally, Gally may be no more infective (defective in terms of infectivity) due to some mutation(s) in tail genes (or Gally might be acquired by LF82 by some mechanism other than normal infection).

P18, L1 (highly): Should be " widely" or " frequently"

P19, L6 (hyperactive): Gally is active, but it can not be said that it is hyperactive.

Reviewer #3: (No Response)

PLOS authors have the option to publish the peer review history of their article (what does this mean?). If published, this will include your full peer review and any attached files.

Reviewer #1: No

Reviewer #2: No

Reviewer #3: No
---

## [Decision Letter · Decision Letter 1]

17 Jan 2023

Dear Dr Lecointe,

We are pleased to inform you that your manuscript 'Phage production is blocked in the adherent-invasive Escherichia coli LF82 upon macrophage infection' has been provisionally accepted for publication in PLOS Pathogens.

Best regards,

William Navarre

Academic Editor

PLOS Pathogens

Raphael Valdivia

Section Editor

PLOS Pathogens

Kasturi Haldar

Editor-in-Chief

PLOS Pathogens

orcid.org/0000-0001-5065-158X

Michael Malim

Editor-in-Chief

PLOS Pathogens

orcid.org/0000-0002-7699-2064

The reviewers feel that the revisions made were sufficient and greatly improved the manuscript. One reviewer did spot some minor typographical issues. Please address these before submitting a final version.

Reviewer Comments (if any, and for reference):

Reviewer's Responses to Questions

**Part I - Summary**

Reviewer #1: This is a revised manuscript on the role of prophage induction in the survival of the AIEC strain LF82 in macrophage. Through the revision, the authors fully addressed my major and minor comments. Therefore, I do not have any more concerns about this manuscript.

Reviewer #2: All the concerns I raised have been nicely addressed, and the manuscript has been improved very well. There are a few minor points to be modified, but the authors can easily modify them.

**Part II – Major Issues: Key Experiments Required for Acceptance**

Reviewer #1: None

Reviewer #2: None

**Part III – Minor Issues: Editorial and Data Presentation Modifications**

Reviewer #1: None

Reviewer #2: S1 Fig: I suppose the heat map and color indications (of genes) used for this figure are the same as those used for Fig 1. Please indicate this point in the legend.

Page 7 Line 6 (Most morons were of unknown function): "Most" is not appropriate here and needs to be reworded.

Page 9 Line30 (TMP protein): this should be "tape measure protein"

Page 10, Line10 (Tab 4): Tab 4 appears earlier in the text than Tab 3 (cited in the M&M section).

Page 13 Lines 13, 1521 (qval): q-value?

Page 17, Line 11 (K12 lysogen): K-12 lambda lysogen?

Page 18, Line32 (receptor strains): recipient strains

PLOS authors have the option to publish the peer review history of their article (what does this mean?). If published, this will include your full peer review and any attached files.

Reviewer #1: No

Reviewer #2: No

---

## [Editor Report · Acceptance letter]

30 Jan 2023

Dear Dr Lecointe,

We are delighted to inform you that your manuscript, "Phage production is blocked in the adherent-invasive *Escherichia coli* LF82 upon macrophage infection ," has been formally accepted for publication in PLOS Pathogens.

Best regards,

Kasturi Haldar

Editor-in-Chief

PLOS Pathogens

orcid.org/0000-0001-5065-158X

Michael Malim

Editor-in-Chief

PLOS Pathogens

orcid.org/0000-0002-7699-2064